# Mitigating Backdoor Attacks in Federated Learning through Noise-Guided Aggregation

## Abstract

Backdoor attack in federated learning (FL) has attracted much attention in the literature due to its destructive power. Advanced backdoor defense methods mainly involve modifying the server's aggregation rule to filter out malicious models through some pre-defined metrics. However, calculating these metrics involves malicious models, leading to biased metrics and defense failure. Therefore, a straightforward approach is to design a metric not tainted by malicious models. For instance, if the server has private data to evaluate model performance, then model performance would be an effective metric for backdoor defense. However, directly introducing data-related information may cause privacy issues. Thus, we propose a *n*oise-gu*i*ded *r*obust *a*ggregation (Nira), which trains and evaluates models using pure noise. Specifically, Nira constructs a noise dataset and shares it across the server and clients, enabling clients to train their models over the shared noise and local data. To ensure the generalizability of models trained on noise, Nira encourages clients to align their local data to shared noise in the representation space. Consequently, Nira can filter out models prior to aggregation according to the model performance, e.g., prediction accuracy on noise. We conduct extensive experiments to verify the efficacy of Nira against backdoor attacks, demonstrating the superiority over previous works by a substantial margin.

## 1 Introduction

Federated Learning (FL) (McMahan et al., 2017; Kairouz et al., 2021) is a powerful learning scheme enabling multiple clients to train a global model collaboratively, without leaking their private information. This decentralized nature provides significant advantages over traditional centralized learning, particularly in applications where data privacy is a concern, such as recommendation (Isinkaye et al., 2015; Wu et al., 2021), computer vision (LeCun et al., 1998; Zhu et al., 2020), and healthcare (Xu et al., 2021; Yuan et al., 2020). Although significant progress has been made, FL is still vulnerable to various security threats, such as adversarial attacks. Therefore, how to enable FL to be adversarially robust remains an open question.

This paper focuses on a particular adversarial attack named *backdoor attack* (Chen et al., 2017; Gu et al., 2019; Liao et al., 2018), which is recognized to be very harmful in FL (Bhagoji et al., 2019; Bagdasaryan et al., 2020; Wang et al., 2020a). In general, backdoor attackers manipulate training data on clients, sending client models trained on such tampered data to the server to pollute the global model. Then, after awakening some trigger embeds (i.e., the backdoor) on new inputs, the global model will predict the designated targets given by the attackers. For example, an attacker can make the global model predict a specific label (e.g., classify blue trucks as birds) when seeing a particular triggered input (e.g., an image of a blue truck with a particular pattern). The backdoor attack is among the most lethal ways of poisoning (Biggio et al., 2012; Liu et al., 2018) and model stealing (Tramèr et al., 2016; Juuti et al., 2019), posing a great threat to the robustness of real-world FL systems. Hence, it is essential to investigate effective methods for backdoor defense in FL.

Many efforts have been devoted to backdoor defense in FL, where advanced methods mainly detect attacks by analyzing some pre-defined metrics of client models, such as Euclidean distance (Blanchard et al., 2017; Pillutla et al., 2022), mean value (Yin et al., 2018), cosine similarity (Fung et al., 2018), and norm bounds (Sun et al., 2019; Panda et al., 2022). However, since malicious models are also involved in the calculation of these metrics, these methods may yield tainted metrics and fail

to achieve effective defense. For example, in scenarios where the proportion of malicious clients is significant, these majority-based defense methods may erroneously categorize and exclude the models of the minority benign clients as attackers, thus rendering poor defense performances.

To address the tainted metric issue, a straightforward approach is to design a metric not tainted by malicious models. To this end, we propose *no*ise-gu*i*ded *r*obust *a*ggregation (Nira), a novel backdoor defense method that can bypass the troubles raised by de-centralization in FL. Overall, Nira provides a surrogate dataset to the server, thereby allowing the model's performance on the surrogate data to become an effective metric. This surrogate dataset is synthesized by pure Gaussian noise data containing no privacy information from clients (thus without privacy leakages). Nira shares the dataset across the server and clients, thus clients can train local models with both the local data and the shared surrogate data. Such a noise surrogate dataset can effectively assist the server in filtering out malicious models by evaluating the prediction accuracy and features of the model on it, as depicted in Figure 1a.

However, simply adding the pure noise data to the training process can potentially impact the performance of local models on the natural data. This is because the original natural data and the noise data have different distributions. To ensure the generalizability of models trained on noise, inspired by the joint distribution alignment (Long et al., 2017), we calibrate the feature distributions of natural data and noise data. Consequently. the noise dataset can represent the training data, enabling the identification of outliers that deviate from the distribution of benign clients. By completing this task, the noise dataset will not hurt the model training of honest clients while helping the server filter out attackers. Figure 1b and Figure 1c demonstrate Nira's defense capabilities. It can be seen that existing methods experience a rapid increase in attack rate as the proportion of attackers exceeds 50%. In contrast, Nira is able to accurately filter out malicious models and keep the attack rate below 1% when the proportion of attackers is less than 70%, demonstrating its efficacy.

Our main contributions can be summarized as follows:

- We point out that metrics used for backdoor defense can be tainted by malicious models, leading to the failure of existing approaches.

- We propose a noise-guided robust aggregation (Nira) mechanism to filter out malicious models, shedding light on backdoor defense. Specifically, Nira introduces a surrogate dataset containing pure noise and leverages the model performance on noise data as a metric to filter out malicious clients.

- Comprehensive experiments demonstrate the effectiveness of Nira. Moreover, we empirically find that Nira is less likely to mistakenly identify benign clients as malicious attackers when no malicious clients are in the federated system, preventing resource wastage and exhibiting marginal performance degeneration.

## 2 Preliminaries

To begin with, we introduce the necessary backgrounds about federated learning 2.1, backdoor attack 2.2, and domain adaptation 2.3.

### 2.1 Federated Learning

The federated learning (FL) process is executed by a set of clients in synchronous update rounds, and the server aggregates the local model updates of selected clients in each round to update the global model. Formally, FL aims to minimize a global objective function: $\min_w F(w) := \sum_{k=1}^{K} p_k F_k(w)$, where $K$ is the number of all clients, $p_k \geq 0$ is the weight of $k$-th client, and $F_k$ is the local objective function: $F_k(w) := \mathbb{E}_{(x,y) \sim \mathcal{D}_k(x,y)} \ell(f(x;w), y)$. We denote $\mathcal{D}_k(x,y)$ as the data distribution in the $k$-th client, $\ell(\cdot, \cdot)$ as the loss function such as cross-entropy, and $f$ as the classifier which consist of a feature extractor $\phi$ and a predictor $\rho$, i.e., $f = \rho \circ \phi$.

At each communication round $t$, the server uniformly selects a subset of clients $\mathcal{S}^t$ from the federated system and sends them the current global model $G^t$. The each selected client $k$ performs $E$ epochs local updates to get a new local model $L_k^t$ by training on their private datasets:

$$L_{k,j+1}^t = L_{k,j}^t - \eta_{k,j} \nabla F_k \left( L_{k,j}^t \right), j \in \{0, 1, \cdots, E-1\}, \tag{1}$$

where $\eta_{k,j}$ is the learning rate, and $L_{k,j}^t$ represents the model after $j$-th updates, i.e., $L_{k,0}^t = G^t$ and $L_{k,E}^t = L_k^t$. Then, all selected clients send the local models back to the server, and the server aggregates these models to produce a global model. Typically, the aggregation is performed using the following sample-based weighting manner (McMahan et al., 2017):

$$G^{t+1} = \sum_{k \in \mathcal{S}^t} \frac{n_k}{\sum_{i \in \mathcal{S}^t} n_i} L_k^t, \qquad (2)$$

where $n_k$ is the number of training samples on the $k$-th clients. In FL, data distributions typically vary with clients, which is known as the Non-IID federated learning setting, posing a client drift challenge.

## 2.2 BACKDOOR ATTACKS

Backdoor attacks aim to manipulate local models to fit both the main task and the backdoor task simultaneously, inducing the global model to behave normally on untampered data samples while achieving a high attack success rate on backdoored data samples. We consider the *strong attacker* (Bagdasaryan et al., 2020) who can fully control the compromised client, including the private local data and the model training process. When there are multiple attackers, we assume they can collude with each other and share the same target. As discussed in (Sun et al., 2019), the participating patterns of attackers can be divided into the *fixed frequency* attack, where the attacker periodically participates in the FL round, and the *random sampling* attack, where the attacker can only perform attacks during the FL rounds in which they are selected. We consider the random sampling case in this paper since this setting is more common in real-life scenarios. The backdoor can also be divided into the *semantic backdoor* (Bagdasaryan et al., 2020; Wang et al., 2020a), which denotes samples that share the same semantic property, and the *trigger-based backdoor* (Xie et al., 2020), which denotes samples that contain a specific "trigger". Here we consider the trigger-based backdoor attacks following previous work(Xie et al., 2020; Zhang et al., 2022). Furthermore, we form the backdoor task by conducting model replacement attacks introduced in (Bagdasaryan et al., 2020).

## 2.3 DOMAIN ADAPTATION

The core challenge in domain adaptation is how to address the impact of the inconsistency between the distribution of training data and testing data(Pan & Yang, 2010), referred to as the source domain and the target domain. Distribution shift can be classified according to the components that cause the shift into covariate shift(Pan et al., 2010; Ben-David et al., 2010), conditional shift(Zhang et al., 2013; Gong et al., 2016), and dataset shift(Quinonero-Candela et al., 2008; Long et al., 2013; Zhang et al., 2020), corresponding shifts for $\mathcal{D}(x)$, $\mathcal{D}(x|y)$ and $\mathcal{D}(x,y)$ respectively.

A commonly used and effective method for reducing the impact of distribution shift is to use a feature extractor $\phi$ to extract similar feature distributions from the source distribution $\mathcal{D}_S$ and the target distribution $\mathcal{D}_T$(Ganin et al., 2016; Zhao et al., 2019; Long et al., 2017). Specifically, the feature extractor $\phi$ minimizes the distribution discrepancy for three types of distribution shift with the measurement $d$ respectively(Ganin et al., 2016; Gong et al., 2016): the marginal distribution discrepancy $d(\mathcal{D}_S(\phi(x)), \mathcal{D}_T(\phi(x)))$, the conditional distribution discrepancy $d(\mathcal{D}_S(\phi(x)|y), \mathcal{D}_T(\phi(x)|y))$ and the joint distribution discrepancy $d(\mathcal{D}_S(\phi(x), y), \mathcal{D}_T(\phi(x), y))$. In this paper, we need to consider the most challenging dataset shift and minimize the joint distribution discrepancy. We regard the noise dataset as the source domain and the original local data distribution as the target domain.

## 3 NOISE-GUIDED ROBUST AGGREGATION

This section proposes a novel noise-guided robust aggregation (Nira) approach to defend against backdoor attacks by introducing a special dataset containing pure noise to assist the server in identifying and filtering malicious clients.

### 3.1 SURROGATE DATASET

The failure of existing methods can be mainly attributed to the tainted metrics used for filtering out malicious clients. Specifically, existing methods for defending against backdoor attacks in FL fo-

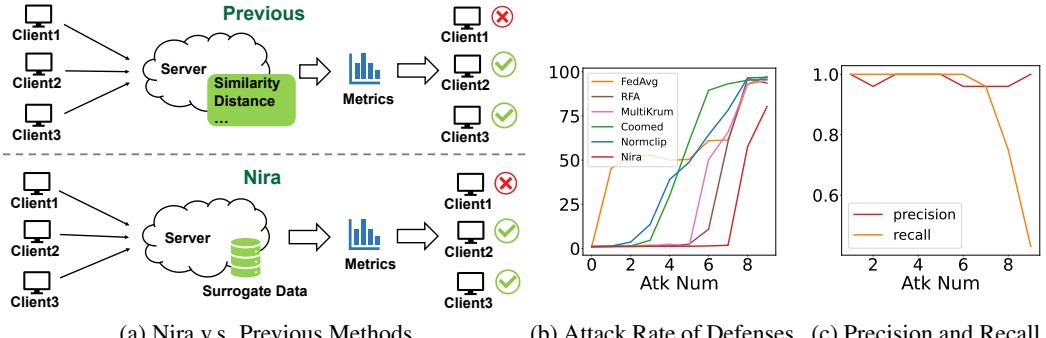

(a) Nira v.s. Previous Methods   (b) Attack Rate of Defenses   (c) Precision and Recall

Figure 1: (a) Illustration of Nira and Previous Methods; (b) Average attack rate of defense methods in the 5 rounds before the best training accuracy on CIFAR-10 with 10 clients in total. All clients are selected to aggregate at every communication round; (c) Nira's precision and recall in filtering out malicious clients.

cus on studying the attributes of the received client models themselves, such as taking the mean or median of the model parameters (Yin et al., 2018), filtering out outliers based on squared-distance of updates (Blanchard et al., 2017), and clipping updates with excessive norms (Sun et al., 2019). Consequently, these methods encounter a dilemma when the proportion of malicious clients is significant: these metrics will become unreliable and render poor defense performances.

A more direct approach to backdoor defense is to provide the server with some typical data and distinguish attacks based on the performance of the client models on these data. However, in order to protect privacy, the client cannot directly share local training data with the server. Therefore, we propose to construct a surrogate dataset that contains no private information. The server sends the surrogate dataset to all clients and requires them to train local models with both the original local data and the shared surrogate data:

$$F_k^{cls} := \mathbb{E}_{(x,y)\sim\mathcal{D}_k}\ell(f(x),y) + \mathbb{E}_{(x,y)\sim\mathcal{D}_n}\ell(f(x),y), \qquad (3)$$

where $\mathcal{D}_n$ is the distribution of the surrogate dataset. This objective function is formulated to ensure that the local client model performs well on the surrogate dataset, serving as a crucial reference for the server to identify potential attackers.

### 3.2 FEATURE DISTRIBUTION ALIGNMENT

Intuitively, simply adding a surrogate dataset that has a completely different distribution from the original local data will harm the model's generalization performance (Frénay & Verleysen, 2013; Polyzotis et al., 2017). Therefore, inspired by the previous work (Long et al., 2017), which investigates the joint distribution discrepancy, we further introduce feature distribution alignment to enable the models trained on surrogate data to perform well on the natural distribution.

To represent real data using surrogate data, it is also crucial to align the distribution of real features with that of surrogate ones. Since the surrogate data cannot contain any private information, we propose that the surrogate dataset can be generated using pure Gaussian noise, such as random noise derived from a randomly initialized StyleGAN (Karras et al., 2019), as depicted in Figure 3. To ensure that the model transfers the knowledge acquired from the surrogate dataset to the real dataset, we draw inspiration from domain adaptation techniques. Specifically, we consider the surrogate dataset as the source domain and the real dataset as the target domain and perform domain adaptation to mitigate the generalization risk of the real distribution. By leveraging the fundamental principles of domain adaptation, we align these two distributions in the feature space and ensure the good performance of the model trained with surrogate data on real data.

Following the previous works on addressing dataset shift (Long et al., 2013; 2017; Lei et al., 2021), we minimize the joint distribution discrepancy between real features and surrogate features. Note that the surrogate dataset is arbitrarily constructed, this allows us to generate appropriate noise data with the same label distribution as the real data. Consequently, we only need to minimize the

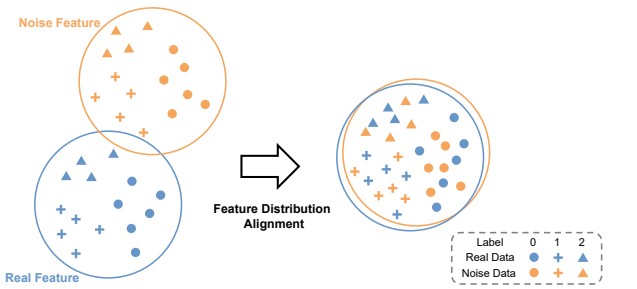

Figure 2: Feature Distribution Alignment

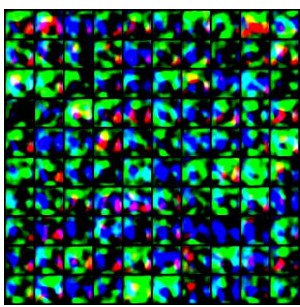

Figure 3: Noise Data

conditional distribution discrepancy rather than the joint distribution discrepancy. In particular, We propose the objective for the feature distribution alignment:

$$F_k^{da} := \mathbb{E}_y d(\mathcal{D}_k(\phi(x)|y), \mathcal{D}_n(\phi(x)|y)), \tag{4}$$

where $\phi$ is the feature extractor that composes the classifier $f = \rho \circ \phi$, $\mathcal{D}_k(\phi(x)|y)$ and $\mathcal{D}_n(\phi(x)|y)$ are the conditional feature distributions obtained by the feature extractor $\phi$ on the real dataset and the surrogate dataset, respectively. The insight of this objective is shown in Figure 2. This objective encourages the feature extractor to learn the same conditional feature distribution from two different data distributions.

Thereafter, we propose the overall objective of the client during local training in the Nira framework:

$$F_k^{Nira} = F_k^{cls} + \lambda F_k^{da}, \tag{5}$$

where $\lambda$ is a hyperparameter that governs the trade-off between classification accuracy on training data and the degree of alignment in feature distribution. Nira enables the model to accurately classify real data and noise data, and simultaneously encourages the model to generate similar features from real data and noise data with the same label, thus achieving good generalization performance on both distributions. Empirical observations in Figure 4 demonstrate this effect.

To theoretically prove the effectiveness of the proposed feature distribution alignment, we analyze the relationship between the model's generalization performance and the distribution shift. Based on the existing theoretical conclusions, the generalization performance is related to the margin between samples and the decision boundary. Therefore, we first introduce the definition of *statistical robustness* between two distributions, serving as a metric for quantifying the degree of generalization performance.

**Definition 3.1** (Statistical Robustness). *We define statistical robustness for a classifier $f$ on a distribution $\mathcal{D}$ according to a distance metric $d$: $SR_d(f, \mathcal{D}) = \mathbb{E}_{(x,y) \sim \mathcal{D}} \inf_{f(x') \neq y} d(x', x)$, where classifier $f : \mathcal{X} \to \mathcal{Y}$ predicts class label of an input sample.*

The defined statistical robustness refers to the expected distance from each sample to the closest adversarial example. Hence, for the model $f$ learned from the source distribution $\mathcal{D}_n(x, y)$, we can quantify the generalization performance on the target distribution $\mathcal{D}(x, y)$. To achieve good generalization performance, we aim to provide a lower bound on the transferred statistical robustness, i.e., $\mathbb{E}_{\substack{\mathcal{S} \sim \mathcal{D}_n \\ f \leftarrow Nog(\mathcal{S})}} SR_d(f, \mathcal{D})$, where $f \leftarrow Nog(\mathcal{S})$ means the model $f$ is trained on the training set $\mathcal{S}$ using Nira. To this end, we have the following theorem.

**Theorem 3.1.** *Let $f$ be a neural network, $\mathcal{D}(x, y)$ and $\mathcal{D}_n(x, y)$ are two separable distributions with identical label distributions, corresponding to the distributions of real data and noise data, respectively. Then, training the model with the proposed objective for the feature distribution alignment, i.e., Eq. 5 elicits the bounded statistical robustness.*

We provide the proof in Appendix A.1. Theorem 3.1 shows that the model trained with the proposed objective can learn to provable generalization performance, which is consistent with the previous work (Long et al., 2013; 2017) that aligning the joint distribution between the source and target domains.

### 3.3 BACKDOOR FILTERING

At each round, the server aggregates the client models trained with their original local dataset and the surrogate noise dataset. Compared to previous methods that merely focused on the characteristics of the model itself, the server can more effectively identify and filter backdoor attackers by evaluating the model's performance on the surrogate dataset, allowing for a more comprehensive assessment of the model's capabilities and can lead to more accurate identification of backdoor threats. Specifically, we filter backdoor attacks through two steps: *accuracy test* and *feature distribution test*.

The server uniformly selects $b$ samples from the surrogate dataset each round. During the accuracy test stage, the server tests the classification accuracy of received $|\mathcal{S}|$ local models on these samples, and filters out models with accuracy lower than a specific threshold $\sigma_1$. This individual evaluation approach renders the metric impervious to variations in the attacker's ratio. We denote the set of models that remain in the first step as $\mathcal{S}_1$. Then during the feature distribution test stage, for each model in the set $\mathcal{S}_1$, the server calculates the squared distance between the features of the model on $b$ noise samples, obtaining a distance matrix $D \in \mathbb{R}^{b \times b}$. The "features of the model" refer to the output features of a specific layer in a neural network. Then for each matrix in $|\mathcal{S}_1|$ distance matrices, the server calculates the average distance between it and the other $|\mathcal{S}_1| - 1$ matrices, and filters out models with the average distance higher than a specific threshold $\sigma_2$. These two steps can be formally represented by the following equations:

$$
\begin{aligned}
\mathcal{S}_1 &= \{L_i \mid \mathrm{acc}\,(L_i) \le \sigma_1, \forall L_i \in \mathcal{S}\}, \\
\mathcal{S}_2 &= \{L_i \mid \mathrm{dis}\,(D_{L_i}) \le \sigma_2, \forall L_i \in \mathcal{S}_1\},
\end{aligned}
\tag{6}
$$

where acc($\cdot$) denotes the accuracy of the model on surrogate data, and dis($\cdot$) denotes the average distance between the distance matrix of the model and those of other models. Here we measure distance using the Frobenius norm. For the models that remain after the second step, the server aggregates them and updates the global model. The server can effectively and precisely filter backdoored models by employing these two steps with the help of the surrogate noise dataset. To efficiently select thresholds, we assume that the server can identify a small number of benign clients and simulate the training process. In Appendix E.5, we introduce a method for efficiently selecting thresholds and also investigate another interval-based filtering strategy to adapt to situations where thresholds are difficult to determine.

### 3.4 OVERVIEW OF NIRA

In the proposed Nira framework, the server crafts a surrogate dataset that consists of pure Gaussian noise generated by an untrained Style-GAN. We show the noise samples in Figure 3. Then all clients receive the surrogate dataset and train local models with the objective Eq. 5. Note that for the client, Nira only modifies its training data and objective function, whereas for the server, Nira only introduces two additional model filtering steps. Therefore, Nira can be combined with most federated learning algorithms, including FedAvg, FedProx (Li et al., 2020) and FedNova (Wang et al., 2020c). The overall procedure of Nira coupled with FedAvg is illustrated in Appendix D. Note that malicious attackers may reject to follow the proposed protocol. In this context, the model sent from malicious attackers will produce a poor performance on the constructed surrogate data. Accordingly, our method will detect these models as malicious models.

## 4 EXPERIMENTS

The goal of our empirical study is to demonstrate the improved defense capability of Nira over the state-of-the-art FL defense methods. We conduct our experiments on image classification tasks over three datasets: CIFAR-10(Krizhevsky et al., 2009), FMNIST(Xiao et al., 2017) and SVHN(Netzer et al., 2011). We simulate FL for $R$ rounds among $K$ clients, of which $m$ are corrupted by attackers. In each round, the server uniformly selects $C \cdot K$ clients for some $C \le 1$ and sends the global model to each selected client. The selected clients then perform local training on the received model for $E$ epochs and send the updates back to the server. The goal of attackers is to make the aggregated model misclassify samples poisoned by triggers into the target class. For the aggregated model on the server, we measure three performance metrics: total accuracy, attack rate and main accuracy.

Total accuracy is computed on the entire test dataset, while attack rate measures the proportion of test samples with triggers classified as the target label by the model, and main accuracy is computed on clean test samples. Our experimental results show that Nira significantly outperforms baseline methods in defending against backdoor attacks.

## 4.1 EXPERIMENT SETUP

**Datasets and Models.** To evaluate the effectiveness of Nira, we conduct experiments on three computer vision datasets including CIFAR-10, FMNIST and SVHN in the FedML framework(He et al., 2020). We use ResNet-18(He et al., 2016) as the shared global model in FL for all three datasets. We utilize the partition method Latent Dirichlet Sampling(Hsu et al., 2019) to partition datasets, generating a local dataset for each client, and using the parameter $\alpha$ to control the degree of Non-IID. We set $\alpha = 1$ to simulate the Non-IID setting by default and conduct experiments under the IID setting in Appendix E.3.

**Surrogate Datasets.** At the beginning of the training phase, an un-pretrained StyleGAN-v2(Karras et al., 2020) is utilized to generate a surrogate dataset without using any training data. The server samples from various Gaussian distributions, each with the same mean but different standard deviations, to generate noise images with diverse latent styles. Each style corresponds to a distinct class. Then the generated noise images are distributed to all clients and used together with the original datasets for local training. The size of the surrogate dataset in our experiments is 2000. We show the surrogate dataset in Appendix E.6.

**Random sampling attack.** The attack model considered in our work is the random sampling attack as discussed in (Sun et al., 2019), where the attackers have complete control over a fraction of clients. In each FL round, the server randomly selects a subset of clients to participate in the training process. The attackers are only able to affect the training of the global model during the rounds in which they are selected. The number of selected attackers in each round follows a hypergeometric distribution.

**Backdoor tasks.** The backdoor task aims to make the global model misclassify backdoored samples into the target class. Since the server randomly selects clients in each round, multiple attackers may be chosen during a single round. We assume that attackers can collude and share the same target, i.e. all attackers aim to make the global model misclassify backdoored samples into the same target class. For the CIFAR-10 and FMNIST datasets, attackers aim to misclassify into class '2', and for the SVHN dataset, attackers aim to misclassify into class '5'. In each round, attackers implant the trigger into partial samples of each class based on the poison ratio, re-label them with the target class, and then train the local model on the backdoored dataset. The trigger we use is a hollow white rectangle implanted in the upper left corner of the poisoned sample. When employing Nira for defense, the noise dataset will be combined with the backdoored original local dataset for training. Attackers further perform model replacement attacks(Bagdasaryan et al., 2020) to generate malicious local models and send them to the server.

**Defense techniques.** We conduct FedAvg(McMahan et al., 2017) as the baseline FL aggregation algorithm. The results using FedProx are reported in Appendix E.2. To demonstrate the effectiveness of Nira in defending against backdoor attacks, we consider five commonly used defense techniques: (i) Krum and (ii) Multi-Krum(Blanchard et al., 2017); (iii) Coordinate-wise median(Coomed)(Yin et al., 2018); (iv) Norm clipping(Normclip)(Sun et al., 2019) and (v) RFA(Pillutla et al., 2022). The detailed hyper-parameters of these algorithms are reported in Appendix C.

## 4.2 EXPERIMENTAL RESULTS

To compare the performance of different defense algorithms, we use three metrics: the average total accuracy (Acc), the average attack success rate (Atk Rate), and the average accuracy of main tasks (Main Acc) in the 5 rounds before the model converges. We conduct FL with a maximum of 200 rounds using the adopted defense algorithm on CIFAR-10, FMNIST and SVHN. There are 50 clients in total, the number of backdoor attackers can range from 0 to 12, and the poison ratio can range from 1% to 20%, depending on different settings. In each round, the server randomly selects 20 clients to participate in training and sends them the global model. The selected clients then perform local training for 1 epoch on the received model and send the locally trained model to the server.

Table 1: Acc, Atk Rate and Main Acc of defense algorithms on CIFAR-10, FMNIST and SVHN when defending against varying attackers. The poison ratio is 5%.

| Atk Num | Defense | CIFAR-10 | | | FMNIST | | | SVHN | | |
|---|---|---|---|---|---|---|---|---|---|---|
| | | Acc | Atk Rate | Main Acc | Acc | Atk Rate | Main Acc | Acc | Atk Rate | Main Acc |
| 0 | FedAvg | 84.72 | 3.26 | 84.87 | 91.51 | 1.77 | 91.64 | 89.2 | 1.12 | 89.28 |
| | RFA | 84.92 | 1.44 | 84.99 | 91.69 | 1.88 | 91.85 | 89.22 | 0.98 | 89.27 |
| | Krum | 50.24 | 3.88 | 49.85 | 86.69 | 3.57 | 86.79 | 79.65 | 1.97 | 79.68 |
| | MultiKrum | 80.77 | 1.9 | 80.81 | 91.14 | 1.86 | 91.27 | 85.89 | 1.68 | 85.97 |
| | Coomed | 83.92 | **1.16** | 83.94 | **91.79** | 1.95 | **91.88** | 88.99 | 1.19 | 89.07 |
| | Normclip | 85.07 | 1.38 | 85.18 | 91.54 | 1.68 | 91.66 | 87.56 | 1.46 | 87.67 |
| | Nira(ours) | **86.29** | 1.33 | **86.28** | 91.2 | **1.38** | 91.26 | **90.32** | **0.82** | **90.39** |
| | Nira Adapt(ours) | 86.29 | 1.33 | 86.28 | 91.2 | 1.38 | 91.26 | 90.32 | 0.82 | 90.39 |
| 4 | FedAvg | 76.79 | 87.63 | 83.4 | 83.71 | 99.78 | **92.09** | 81.41 | 67.14 | 86.66 |
| | RFA | 79.38 | 76.31 | 85.2 | 87.41 | 22.74 | 89.2 | 88.99 | 5.36 | 89.36 |
| | Krum | 49.58 | 5.5 | 49.41 | 84.12 | 5.3 | 84.23 | 83.51 | 1.26 | 83.55 |
| | MultiKrum | 80.5 | 3.24 | 80.67 | 91.23 | 1.99 | 91.36 | 86.47 | 1.37 | 86.53 |
| | Coomed | 76.75 | 58.44 | 80.9 | 88.93 | 5.83 | 89.36 | 89.04 | 3.09 | 89.25 |
| | Normclip | 79.41 | 78.17 | **85.41** | 88.94 | 8.55 | 89.58 | 87.58 | 2.19 | 87.7 |
| | Nira(ours) | **85.2** | **2.3** | 85.31 | **91.46** | **1.44** | 91.55 | 90.72 | 1 | 90.75 |
| | Nira Adapt(ours) | 84.28 | 2.41 | 84.4 | 90.22 | 1.57 | 90.33 | **91.12** | **0.89** | **91.19** |
| 8 | FedAvg | 77.86 | 88.9 | 84.67 | 83.27 | 99.67 | **91.61** | 80.79 | 66.21 | 85.9 |
| | RFA | 78.69 | 83.42 | 85.07 | 84.43 | 30.17 | 86.8 | 87.59 | 4.65 | 87.88 |
| | Krum | 50.91 | 6.44 | 50.79 | 81.87 | 3.6 | 82 | 80.55 | 1.36 | 80.55 |
| | MultiKrum | 78.73 | 6.28 | 79 | 89.42 | 2.93 | 89.55 | 86.58 | 1.4 | 86.63 |
| | Coomed | 77.27 | 81.88 | 83.42 | 84.68 | 10.17 | 85.49 | 86.83 | 7.05 | 87.25 |
| | Normclip | 78.85 | 83.33 | 85.26 | 86.9 | 7.78 | 87.5 | 87.34 | 5.16 | 87.69 |
| | Nira(ours) | **85.25** | **2.49** | **85.42** | 90.79 | **1.82** | 90.89 | **90.68** | **0.88** | **90.72** |
| | Nira Adapt(ours) | 84.93 | 2.52 | 85.02 | **90.83** | 1.9 | 90.87 | 90.58 | 1.17 | 90.65 |
| 12 | FedAvg | 77.8 | 89.64 | 84.67 | 83.34 | 99.64 | **91.68** | 81.58 | 75.29 | 87.61 |
| | RFA | 77.9 | 87.76 | 84.61 | 81.88 | 44.36 | 85.32 | 86.13 | 8.12 | 86.64 |
| | Krum | 48.03 | 8.47 | 47.9 | 80.02 | 10.89 | 80.51 | 80.74 | 2.6 | 80.8 |
| | MultiKrum | 76.99 | 10.98 | 77.72 | 86.63 | 13.94 | 87.71 | 86.77 | 1.57 | 86.83 |
| | Coomed | 77.12 | 84.62 | 83.49 | 83.31 | 26.27 | 85.34 | 86.47 | 13.57 | 87.39 |
| | Normclip | 78.68 | 87.91 | **85.47** | 85.24 | 15.08 | 86.4 | 86.31 | 13.39 | 87.18 |
| | Nira(ours) | **84.68** | **2.87** | 84.9 | **87.29** | **3.07** | 87.38 | 89.48 | **1.44** | 89.52 |
| | Nira Adapt(ours) | 84.22 | 2.97 | 84.44 | 87.1 | 3.24 | 87.19 | **89.88** | 1.69 | **89.95** |

**Different Numbers of Attackers.** As shown in Table 1, Nira outperforms other baselines in almost all scenarios when defending against varying numbers of attackers across the three datasets. It can be seen that Nira can improve model performance even without attackers. We conjecture that this is mainly due to several factors: 1) Adding the surrogate dataset reduces data heterogeneity between clients and mitigates client drift; 2) Aligning real feature distribution with the shared noise feature distribution further mitigates client drift; 3) After adding the surrogate dataset, the clients' training data contains more classes, which can alleviate the negative impact caused by the imbalance of sample quantities among different classes.

When there are attackers in FL, Nira's performance is also superior to other baselines. Nira achieves a significantly lower Atk Rate than other baselines while maintaining high model accuracy. In particular, when the number of attackers is 12, Nira reduces the Atk Rate by up to 7.82% compared to the second-ranked method. We notice that the main accuracy of Nira is sometimes slightly lower than the best result. We speculate that this is mainly because Nira filters out malicious models before aggregation, reducing the number of models aggregated. Consequently, the aggregated model becomes more difficult to converge, especially in Non-IID settings. In Appendix E.10, we report the changing curves of Atk Rate, Acc, and attacker filtering rate over rounds.

**Adaptive Attack Scenario.** To further illustrate Nira's defensive capabilities, we assume that the attackers have knowledge about Nira and adopt more specialized attack methods. Specifically, we consider the following adaptive attack scenario: the attackers divide their poisoned dataset into poisoned and benign parts. For the alignment part of the loss, it only aligns the surrogate samples with the data within the benign parts. The results of this adaptive attack are shown in Table 1, denoted as Nira Adapt. It can be seen that Nira still performs relatively well against this adaptive attack. Although Nira's performance is slightly worse under this adaptive attack, it is still better than other defense methods. We conduct experiments under more adaptive attack scenarios in Appendix E.9.

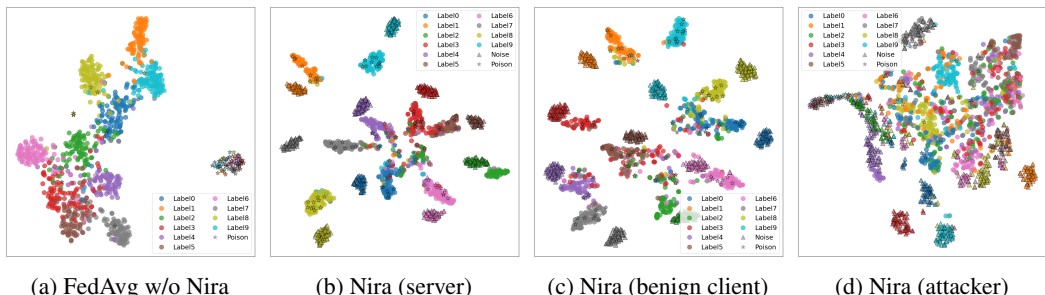

| (a) FedAvg w/o Nira | (b) Nira (server) | (c) Nira (benign client) | (d) Nira (attacker) |

Figure 4: The visualization of the feature distribution of FedAvg with (without) Nira, at the 199-th communication round. The dots represent real data, the triangles represent noise data, the stars represent backdoored data, and different colors indicate different classes.

**Visualization of Feature Distribution.** We exploit t-SNE (Van der Maaten & Hinton, 2008) to visualize the feature distribution, further illustrating how Nira utilizes the noise dataset to help servers defend against backdoor attacks. Specifically, we demonstrate the feature distributions of FedAvg with (without) Nira on the test data for 199 rounds, showcasing their respective generalization capabilities. Figure 4a shows the feature distribution of FedAvg at round 199. It can be observed that FedAvg brings the features from the same class closer together, thereby enabling the classification of different class samples. Meanwhile, the features of samples implanted with triggers are also be clustered, causing the model to misclassify them as the target class. Figure 4b, 4c and 4d represent the feature distributions of Nira for 199 rounds. It can be observed that the attacker's feature distribution is more scattered and loose compared to the benign client, making it perform worse on the surrogate noise data. After filtering out malicious models, the server's features from the same class become closer and more compact, while features of different classes are more distinct, allowing for more accurate classification and improved generalization. The poisoned samples are also correctly clustered together with samples of the same class. We provide more visualization results in Appendix F.

**More Experimental Results.** In order to further investigate the effectiveness, applicability, and scalability of Nira, we conduct more ablation experiments, including changing the poison ratio, coupling Nira with FedProx, conducting experiments in the IID setting, altering the noise generation method, replacing the shared global model, investigating adaptive attack scenarios, and more ablation experiments. We report these experimental results in Appendix E.

### 4.3 LIMITATIONS

The proposed Nira has several limitations that require further investigation. Although our method achieves significant improvement in the experiment, it also introduces additional communication overhead. To mitigate this overhead, we hope to minimize the noise dataset as much as possible. However, the size of the noise dataset may also affect the performance of the client model and the server's ability to detect backdoor attacks accurately. Therefore, future research should investigate the optimal size of the noise dataset that strikes a balance between communication overhead and model performance. This endeavor will enable the development of an efficient and robust model capable of defending against potential attacks.

## 5 CONCLUSION

In this paper, we introduce a generated noise dataset that does not contain real data information into the defense against backdoor attacks in FL. These surrogate noise data provide a more direct and accurate metric for the server to detect malicious models uploaded by backdoor attackers. Through the conditional feature distribution alignment on the noise dataset, our proposed Nira can effectively filter malicious models on the server with the assistance of noise data, without affecting the generalization performance of the local model trained by benign clients. Our empirical results demonstrate that Nira can effectively defend against backdoor attacks and improve the performance of aggregated

models, especially when the proportion of malicious clients is significant, providing new insights for defending against attacks in FL. We hope that our work will inspire further research in developing effective defense mechanisms for FL and contribute to the broader goal of securing machine learning systems.

## ETHIC STATEMENT

This paper does not raise any ethical concerns. This study does not involve any human subjects, practices to data set releases, potentially harmful insights, methodologies and applications, potential conflicts of interest and sponsorship, discrimination/bias/fairness concerns, privacy and security issues, legal compliance, and research integrity issues.

## REPRODUCIBILITY STATEMENT

To make all experiments reproducible, we have listed all detailed hyper-parameters of each FL algorithm. Due to privacy concerns, we will upload the anonymous link of source codes and instructions during the discussion phase to make it only visible to reviewers.

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

# A  PROOF

## A.1  PROOF FOR THEOREM 3.1

*Proof.* We decompose the statistical robustness $SR_d(f, \mathcal{D}(x, y))$ to three quantities as follows:

$$SR_d(f, \mathcal{D}) = (SR_d(f, \mathcal{D}) - SR_d(f, \mathcal{D}_n)) + (SR_d(f, \mathcal{D}_n) - SR_d(f, \tilde{\mathcal{D}}_n)) + SR_d(f, \tilde{\mathcal{D}}_n), \quad (7)$$

where $\tilde{\mathcal{D}}_n$ denotes the empirical distribution for the training set sampled from the noise data distribution $\mathcal{D}_n$. Then based on the linearity of expectation and triangle inequality, we can bound the transferred statistical robustness as follows:

$$\begin{aligned} \mathbb{E}_{f \leftarrow \mathcal{D}} SR_d(f, \mathcal{D}) \geq & \mathbb{E}_{f \leftarrow \mathcal{D}} SR_d(f, \tilde{\mathcal{D}}_n) - |\mathbb{E}_{f \leftarrow \mathcal{D}}[SR_d(f, \mathcal{D}_n) - SR_d(f, \tilde{\mathcal{D}}_n)]| \\ & - |\mathbb{E}_{f \leftarrow \mathcal{D}}[SR_d(f, \mathcal{D}) - SR_d(f, \mathcal{D}_n)]|, \end{aligned} \quad (8)$$

where $\mathbb{E}_{f \leftarrow \mathcal{D}}$ denotes $\mathbb{E}_{\substack{S \sim \mathcal{D} \\ f \leftarrow Nog(S)}}$ for brevity. The three terms above represent the empirical robustness, the generalization penalty and the distribution shift penalty, respectively. Since our goal is to bound the transferred statistical robustness, we need to bound both the generalization penalty and the distribution shift penalty. There are already multiple works (Diochnos et al., 2018; Schmidt et al., 2018; Montasser et al., 2019) have studied the bound of the generalization penalty. In order to bound the distribution shift penalty, we introduce the following lemma:

**Lemma 1.** *Let $\mathcal{D}$ and $\mathcal{D}_n$ be two distributions with identical label distributions, $d(\cdot, \cdot)$ be the Wasserstein distance of two distributions. Then for any classifier $f$, we have:*

$$|SR_d(f, \mathcal{D}) - SR_d(f, \mathcal{D}_n)| \leq \mathbb{E}_y d(\mathcal{D}|y, \mathcal{D}_n|y). \quad (9)$$

We prove Lemma 1 in the follow-up section, i.e., Appendix A.2. With Eq. 8 and Lemma 1, we can further bound the transferred statistical robustness as follows:

$$\begin{aligned} \mathbb{E}_{f \leftarrow \mathcal{D}} SR_d(f, \mathcal{D}) \geq & \mathbb{E}_{f \leftarrow \mathcal{D}} SR_d(f, \tilde{\mathcal{D}}_n) - |\mathbb{E}_{f \leftarrow \mathcal{D}}[SR_d(f, \mathcal{D}_n) - SR_d(f, \tilde{\mathcal{D}}_n)]| \\ & - \mathbb{E}_y d(\mathcal{D}|y, \mathcal{D}_n|y). \end{aligned} \quad (10)$$

The last term $\mathbb{E}_y d(\mathcal{D}|y, \mathcal{D}_n|y)$ is bounded by the proposed objective, i.e., Eq. 5. Thus, the transferred statistical robustness is bounded and the proof is complete.

$\square$

## A.2  PROOF FOR LEMMA 1

*Proof.* To begin with, since the distance metric $d(\cdot, \cdot)$ is the Wasserstein distance, we have:

$$d(\mathcal{D}|y, \mathcal{D}_n|y) = \inf_{J \in \mathcal{J}(\mathcal{D}|y, \mathcal{D}_n|y)} \mathbb{E}_{(x,x') \sim J} m(x, x'), \quad (11)$$

where $\mathcal{J}(\mathcal{D}|y, \mathcal{D}_n|y)$ is the set of joint distributions. Let $\mathcal{J}^*$ be the optimal transport between $\mathcal{D}|y$ and $\mathcal{D}_n|y$. Then we have:

$$
\begin{aligned}
SR_d(f, \mathcal{D}(x,y)) &= \mathbb{E}_{(x,y)\sim\mathcal{D}} \inf_{f(x')\neq y} m\left(x',x\right) \\
&= \mathbb{E}_y \mathbb{E}_{x\sim\mathcal{D}|y} \inf_{f(x')\neq y} m\left(x',x\right) \\
&= \mathbb{E}_y \mathbb{E}_{(x,x'')\sim\mathcal{J}^*} \inf_{f(x')\neq y} m\left(x',x\right) \\
&\leq \mathbb{E}_y \mathbb{E}_{(x,x'')\sim\mathcal{J}^*} \inf_{f(x')\neq y} \left[m\left(x',x''\right) + m\left(x'',x\right)\right] \\
&= \mathbb{E}_y \mathbb{E}_{x''\sim\mathcal{D}_n|y} \inf_{f(x')\neq y} m\left(x'',x'\right) + \mathbb{E}_y \mathbb{E}_{(x,x'')\sim\mathcal{J}^*} m\left(x'',x\right) \\
&= \mathbb{E}_{(x'',y)\sim\mathcal{D}_n} \inf_{f(x')\neq y} m\left(x'',x'\right) + \mathbb{E}_y d\left(\mathcal{D}|y, \mathcal{D}_n|y\right) \\
&= SR_d\left(f, \mathcal{D}_n(x,y)\right) + \mathbb{E}_y d\left(\mathcal{D}|y, \mathcal{D}_n|y\right).
\end{aligned}
\tag{12}
$$

Similarly, we can also prove that:

$$
SR_d(f, \mathcal{D}_n(x,y)) \leq SR_d\left(f, \mathcal{D}(x,y)\right) + \mathbb{E}_y d\left(\mathcal{D}|y, \mathcal{D}_n|y\right). \tag{13}
$$

Now using Eq. 12 and 13 we have:

$$
-\mathbb{E}_y d\left(\mathcal{D}|y, \mathcal{D}_n|y\right) \leq SR_d(f, \mathcal{D}) - SR_d\left(f, \mathcal{D}_n\right) \leq \mathbb{E}_y d\left(\mathcal{D}|y, \mathcal{D}_n|y\right). \tag{14}
$$

Thus, we complete the proof.

$\square$

# B RELATED WORKS

**Federated Learning.** FL is first proposed by (McMahan et al., 2017) to protect data privacy in distributed machine learning. Training models within the FL framework can effectively safeguard privacy, as local data need not be shared. Instead of aggregating local data, the server aggregates local model updates from selected clients to update the global model in each round. To address specific problems within FL, various optimization algorithms have been proposed. FedCurv (Shoham et al., 2019) tackles the catastrophic forgetting problem of FL in the Non-IID case by drawing an analogy with lifelong learning. FedMA (Wang et al., 2020b) reduces the overall communication burden by constructing the global model in a layer-wise manner, matching and averaging hidden elements. There are also many algorithms proposed to address the issue of client drift (Li et al., 2020; Wang et al., 2020c; Karimireddy et al., 2020; Tang et al., 2022), such as FedNova (Wang et al., 2020c), which utilizes normalized averaging to eliminate objective inconsistency. VHL (Tang et al., 2022) also introduces surrogate data into FL, but they focus on solving data heterogeneity issues, while we focus on addressing backdoor attacks.

**Backdoor Attack on Federated Learning.** The goal of backdoor attacks is to modify the global model so that it can produce the desired target labels for inputs that possess specific properties (Shejwalkar et al., 2022). (Bagdasaryan et al., 2020) investigates semantic backdoor attacks where the global model misclassifies input samples with the same semantic property, e.g. misclassifies the blue truck as a bird, and proposes a model-replacement attack that scales up a malicious local update to replace the global model. (Bhagoji et al., 2019) discusses model poisoning attacks launched by a single malicious client. They boost the malicious updates to overcome the impact of updates from benign clients, and further propose alternating minimization and estimating benign updates to evade detection in almost every round. (Wang et al., 2020a) proposes a new category of backdoor attacks called edge-case backdoors, and explains how these edge-case backdoors can lead to detection failures. (Zhang et al., 2022) inserts more durable backdoors into FL systems by attacking parameters that are changed less in magnitude during training. Different from these works that only consider the centralized backdoor attack on FL, (Xie et al., 2020) investigates the distributed backdoor attack (DBA), which decomposes a global trigger pattern into separate local patterns and embeds them into the training set of different adversarial parties respectively.

**Robust Federated Learning.** The goal of robust federated learning is to mitigate the impact of specific attacks during training. (Blanchard et al., 2017) select model update(s) with the minimum squared distance to the updates of other clients. Coordinate-wise median (Yin et al., 2018) selects the median element coordinate-wise among the model updates of clients. Norm clipping (Sun et al., 2019) clips model updates whose norm exceeds a specific threshold. RFA (Pillutla et al., 2022) replaces the weighted arithmetic mean in FedAvg with a weighted geometric median, which is computed using the *smoothed Weiszfeld's algorithm*. FoolsGold (Fung et al., 2018) sums up the historical update vectors and calculates the cosine similarity between all participants to assign a global learning rate to each party. By giving lower learning rates to similar update vectors, Fools-Gold defends against label flipping and centralized backdoor attacks. SparseFed (Panda et al., 2022) utilizes global model top-k sparse updates and client-level gradient clipping to mitigate the impact of poisoning attacks. Our evaluation includes comparisons to five commonly used defense algorithms and demonstrates the stronger capabilities of Nira against backdoor attacks.

## C  IMPLEMENTATION DETAILS

**Model replacement attack**: We form the backdoor task by conducting model replacement attacks (Bagdasaryan et al., 2020). In particular, the attacker trains the local model on the backdoored dataset and gets a backdoored model $X$. The attacker can arbitrarily manipulate the learning rate or training epochs to maximize the attack success rate on the backdoored data. In order to substitute the new global $G^{t+1}$ with the backdoored model $X$, the attacker scales up the weights of $X$ before sending it to the server:

$$L_{atk}^t = \gamma(X - G^t) + G^t,$$

where $\gamma$ is the scaling factor for the balance between attack capability and stealthiness. The scale factor we used is $\frac{m}{n_a}$, where $m$ is the number of clients participating in aggregation each round and $n_a$ is the number of attackers, which is consistent with previous outstanding works (Bagdasaryan et al., 2020; Wang et al., 2020a).

**Krum and Multi-Krum** (Blanchard et al., 2017): Given $n$ clients, Krum aims to defend against a maximum of $f$ attackers. In each round $r$, the server receives $n$ updates $(V_1^r, \cdots, V_n^r)$. For each update $V_i^r$, we denote $i \to j$ as the set of $n - f - 2$ closest updates to $V_i^r$. Then the score for each client $i$ is defined as the sum of squared distances between $V_i$ and each update $V_j$ in the set $i \to j$: $score(i) = \sum_{i \to j} \|V_i - V_j\|^2$. Krum then selects $V_{krum} = V_{i_*}$ with the lowest score $score(i_*) \leq score(i)$ for all $i$, and updates the global model as $w^{r+1} = w^r - V_{krum}$. While Multi-Krum selects $m \in \{1, \cdots, n\}$ updates $V_1^*, \cdots, V_m^*$ with the lowest scores, and calculates their average $\frac{1}{m} \sum_i V_i^*$ to replace $V_{krum}$. In our experiments, we apply $f = 6$ for both Krum and Multi-Krum and set $m = 8$ for Multi-Krum.

**Coordinate-wise median** (Yin et al., 2018): Given the set of updates $(V_1^r, \cdots, V_n^r)$ in each round, Coomed aggregates the updates: $\overline{V}^r = \text{Coomed}\{V_i^r : i \in [n]\}$, where the $j^{th}$ coordinate of $\overline{V}^r$ is given by $\overline{V}^r(j) = \text{med}\{V_i^r(j) : i \in [n]\}$. Here, the function med represents the 1-dimensional median, and $[n] = \{1, \cdots, n\}$.

**Norm clipping** (Sun et al., 2019): Due to the assumption that adversarial attacks can potentially generate updates with large norms, Normclip simply clips model updates whose norm exceeds a specific threshold $M$:

$$w_k^r = \frac{w_k^r}{max(1, \|w_k^r\|_2/M)}.$$

In our experiments, we set the threshold $M = 200$.

**RFA** (Pillutla et al., 2022): RFA replaces the weighted arithmetic mean utilized in FedAvg with a weighted geometric median:

$$\arg\min_v \sum_i \alpha_i \|v - w_i\|,$$

which is computed using the *smoothed Weiszfeld's algorithm*. The weight $\alpha_i$ is set to the proportion of training samples in the client $\alpha_i = \frac{n_i}{\sum_{j \in \mathcal{S}^r} n_j}$, where $\mathcal{S}^r$ is the subset of selected clients at round $r$. For iteration budget $R$ and the parameter $\nu$ in the smoothed Weiszfeld's algorithm, we set $R = 4$ and $\nu = 10^{-5}$.

---

**Algorithm 1** Noise-Guided Robust Federated Learning (Nira)

---

**Input:** local epochs $E$, client number $K$, maximum round $R$, initial parameter $w^0$
**Output:** global parameter $w$
    **Initialization:** Server generates the surrogate noise dataset $\tilde{D}$, and distributes the initial model $w^0$ and $\tilde{D}$ to all clients.

    **Server:**
    **for** each round $r \in \{0, 1, \cdots, R\}$ **do**
        Uniformly selects a subset of clients $\mathcal{S}^r \subseteq \{1, \cdots, K\}$
        Sends the global model $w^r$ to all selected clients $k \in \mathcal{S}^r$
        **for** each client $k \in \mathcal{S}^r$ **in parallel do**
            $w_k^r \leftarrow \text{ClientTraining}(k, w^r)$
        **end for**
        $\mathcal{W}^r \leftarrow \{w_k^r | k \in \mathcal{S}^r\}$
        //Accuracy test
        $\mathcal{W}_1^r \leftarrow \text{AccTest}(\mathcal{W}^r, \tilde{D})$
        //Feature distribution test
        $\mathcal{W}_2^r \leftarrow \text{FeaTest}(\mathcal{W}_1^r, \tilde{D})$
        //Aggregate
        $w^{r+1} \leftarrow \sum p_k w_k^r, w_k^r \in \mathcal{W}_2^r$
    **end for**

    **Benign Client:**
    **for** each epoch $e \in \{0, \cdots, E-1\}$ **do**
        $w_{k,e+1}^r \leftarrow w_{k,e}^r - \eta_{k,e} \nabla F_k^{Nira}\left(w_{k,e}^r\right)$
    **end for**
    Return $w_k^r$ to sever

    **Compromised Client:**
    Injects the backdoor into the local dataset
    **for** each epoch $e \in \{0, \cdots, E-1\}$ **do**
        $w_{k,e+1}^r \leftarrow w_{k,e}^r - \eta_{k,e} \nabla F_k^{Nira}\left(w_{k,e}^r\right)$
    **end for**
    //Scale up the weight
    $w_{atk}^r \leftarrow \gamma(w_k^r - w^r) + w^r$
    Return $w_{atk}^r$ to sever

---

**Nira**: Nira generates the surrogate noise dataset using an un-pretrained StyleGAN-v2 (Karras et al., 2020). Clients then proceed to train local models with the Nira objective parameter $\lambda$ set to 1, and the batch size is set to 128 for both real data and noise data. Then the server receives local models and employs a two-step process to filter out backdoor attackers. For the accuracy test step, we design an accuracy threshold that increases as the training rounds progress: $\sigma_1 = \{\text{round}\,0 : 0, \text{round}\,1 : 0.1, \text{round}\,2 : 0.2, \text{round}\,3 : 0.3, \text{round}\,5 : 0.5, \text{round}\,15 : 0.6, \text{round}\,30 : 0.7, \text{round}\,60 : 0.85, \text{round}\,120 : 0.9, \text{round}\,180 : 0.95\}$. And for the feature distribution test step, we design a distance threshold that decreases as the training rounds advance: $\sigma_2 = \{\text{round}\,0 : 2000, \text{round}\,15 : 1500, \text{round}\,40 : 1300, \text{round}\,70 : 1100, \text{round}\,100 : 1000, \text{round}\,140 : 900\}$. We set the number of noise samples used for evaluating local models $b$ to a fixed value of 128 for all experiments.

For all experiments, the learning rates are set to 0.01 and the learning rate decay is set to 0.992 per round. We employ momentum-SGD as optimizers, with momentum of 0.9 and weight decay of 0.0001. The degree of Non-IID local data distribution on the client is set to $\alpha = 1$.

Table 2: Acc, Atk Rate and Main Acc of defense algorithms on the dataset CIFAR-10 with different poison ratios.

| Poison Ratio | Defense | 4 attackers | | | 8 attackers | | | 12 attackers | | |
|---|---|---|---|---|---|---|---|---|---|---|
| | | Acc | Atk Rate | Main Acc | Acc | Atk Rate | Main Acc | Acc | Atk Rate | Main Acc |
| 1% | FedAvg | 81.12 | 2.41 | 81.21 | 84.43 | 2.41 | 84.51 | 84.97 | 2.76 | 85.07 |
| | RFA | 84.62 | 1.55 | 84.72 | 85.37 | 1.77 | **85.47** | 85.13 | 1.79 | 85.22 |
| | Krum | 49.58 | 5.5 | 49.41 | 51.53 | 1.86 | 51.16 | 52.09 | 3.88 | 51.87 |
| | MultiKrum | 80.5 | 3.24 | 80.67 | 80.49 | 2.3 | 80.66 | 80.49 | 1.53 | 80.5 |
| | Coomed | 83.76 | 1.42 | 83.73 | 83.51 | 1.29 | 83.49 | 83.49 | 1.42 | 83.57 |
| | Normclip | 85.29 | 1.53 | 85.26 | **85.41** | 1.46 | 85.45 | 85.3 | 1.77 | **85.41** |
| | Nira(ours) | **85.59** | **1.2** | **85.71** | 84.89 | **1.27** | 85.03 | 85.14 | **1.35** | 85.18 |
| | Nira Adapt(ours) | 85.59 | 1.2 | 85.71 | 84.54 | 1.71 | 84.64 | 84.38 | 1.49 | 84.43 |
| 5% | FedAvg | 76.79 | 87.63 | 83.4 | 77.86 | 88.9 | 84.67 | 77.8 | 89.64 | 84.67 |
| | RFA | 79.38 | 76.31 | 85.2 | 78.69 | 83.42 | 85.07 | 77.9 | 87.76 | 84.61 |
| | Krum | 49.58 | 5.5 | 49.41 | 50.91 | 6.44 | 50.79 | 48.03 | 8.47 | 47.9 |
| | MultiKrum | 80.5 | 3.24 | 80.67 | 78.73 | 6.28 | 79 | 76.99 | 10.98 | 77.72 |
| | Coomed | 76.75 | 58.44 | 80.9 | 77.27 | 81.88 | 83.42 | 77.12 | 84.62 | 83.49 |
| | Normclip | 79.41 | 78.17 | **85.41** | 78.85 | 83.33 | 85.26 | 78.68 | 87.91 | **85.47** |
| | Nira(ours) | **85.2** | **2.3** | 85.31 | **85.25** | **2.49** | **85.42** | **84.68** | **2.87** | 84.9 |
| | Nira Adapt(ours) | 84.28 | 2.41 | 84.4 | 84.93 | 2.52 | 85.02 | 84.22 | 2.97 | 84.44 |
| 10% | FedAvg | 76.62 | 92.21 | 83.61 | 78.09 | 87.93 | 84.94 | 77.88 | 91.84 | 84.94 |
| | RFA | 78.44 | 84.71 | 84.95 | 78.09 | 88.96 | 84.94 | 78.18 | 91.31 | 85.22 |
| | Krum | 49.58 | 5.5 | 49.41 | 51.89 | 3.83 | 51.55 | 51.64 | 2.03 | 51.28 |
| | MultiKrum | 80.5 | 3.24 | 80.67 | 79.18 | 3 | 79.26 | 74.45 | 50.3 | 77.79 |
| | Coomed | 77.51 | 81.64 | 83.67 | 76.74 | 86.9 | 83.28 | 77.11 | 88.99 | 83.85 |
| | Normclip | 79.04 | 86.05 | **85.7** | 78.58 | 87.52 | **85.35** | 78.55 | 90.61 | **85.55** |
| | Nira(ours) | **85.38** | **2.06** | 85.52 | **85.07** | **2.03** | 85.2 | **84.79** | **2.03** | 84.87 |
| | Nira Adapt(ours) | 85.18 | 2.96 | 85.32 | 83.78 | 2.3 | 84.01 | 84.75 | 2.25 | 84.88 |
| 20% | FedAvg | 73.65 | 91.92 | 80.34 | 78.21 | 90.72 | 85.23 | 76.86 | 92.58 | 83.89 |
| | RFA | 76.47 | 87.71 | 83.08 | 77.83 | 85.65 | 84.39 | 75.62 | 85.46 | 82.01 |
| | Krum | 49.58 | 5.5 | 49.41 | 52.07 | 2.3 | 51.63 | 46.69 | 3.2 | 46.5 |
| | MultiKrum | 80.5 | 3.24 | 80.67 | 79.27 | 1.6 | 79.36 | 75.14 | 68.37 | 79.98 |
| | Coomed | 77.03 | 86.31 | 83.54 | 76.65 | 88.77 | 83.31 | 76.87 | 89.84 | 83.68 |
| | Normclip | 78.62 | 88.81 | 85.49 | 78.7 | 89.05 | **85.62** | 78.32 | 91 | **85.37** |
| | Nira(ours) | 85.53 | **2.01** | 85.71 | **84.92** | **1.51** | 84.97 | **84.28** | **2.01** | 84.42 |
| | Nira Adapt(ours) | **85.78** | 2.34 | **85.85** | 83.97 | 2.36 | 84.07 | 84.27 | 2.08 | 84.41 |

# D ALGORITHM

In Nira, the server generates the surrogate noise dataset at the beginning of the training phase and distributes the noise data to all clients. The clients proceed to train their respective local models using both the original dataset and the noise dataset, and send the trained local models back to the server. To effectively identify and mitigate backdoor attackers, the server employs a two-step filtering process, leveraging the presence of the noise data. We summarize the overall training procedure of Nira in Algorithm 1.

In previous methods, malicious models are also involved in the calculation of these metrics, thus may yield tainted metrics and fail to achieve effective defense. For example, Krum (Blanchard et al., 2017) calculates the sum of the squared distances between each client model and the other client models as its score, and aggregates several models with the lowest scores. However, when the majority are attackers, the score of the malicious model may be relatively lower, leading the server to aggregate malicious models and resulting in defense failure. The proposed method aims to defend against backdoor attacks by designing a metric that will not be tainted by malicious models. To this end, we propose a metric that performs individual evaluation for each local model using surrogate data. This individual evaluation approach renders the metric impervious to variations in the attacker's ratio.

# E MORE EXPERIMENTAL RESULTS

## E.1 DIFFERENT DATASETS AND RATIO

As shown in Table 2 and 4, we conduct experiments on 3 datasets with different poison ratios, ranging from 1% to 20%. The number of attackers is 4. It can be seen that Nira can effectively defend against backdoor attacks under different poison ratios. Note that although the accuracy of

Table 3: Performance of different defense algorithms with FedProx on CIFAR-10.

| Defense | Atk Num = 0 | | | Atk Num = 4 | | | Atk Num = 8 | | | Atk Num = 12 | | |
|---|---|---|---|---|---|---|---|---|---|---|---|---|
| | Acc | Atk Rate | Main Acc | Acc | Atk Rate | Main Acc | Acc | Atk Rate | Main Acc | Acc | Atk Rate | Main Acc |
| FedAvg | 84.92 | 1.97 | 85.03 | 75.47 | 89.56 | 82.13 | 77.61 | 88.9 | 84.39 | 78.34 | 88.9 | **85.21** |
| RFA | 84.27 | 2.23 | 84.4 | 79.46 | 77.67 | 85.36 | 78.71 | 83.26 | **85.08** | 77.78 | 88.07 | 84.5 |
| Krum | 54.54 | 2.82 | 54.25 | 54.64 | 4.34 | 54.51 | 53.53 | 6.44 | 53.45 | 53.41 | 8.17 | 53.38 |
| MultiKrum | 80.71 | 1.73 | 80.84 | 79.58 | 2.03 | 79.66 | 78.87 | 2.22 | 79.04 | 78.21 | 28.44 | 80.23 |
| Coomed | 84.04 | 1.44 | 84.06 | 78.14 | 65.96 | 82.96 | 77.38 | 81.71 | 83.49 | 77.18 | 85.41 | 83.61 |
| Normclip | 84.71 | **1.42** | 84.72 | 78.71 | 77.19 | 84.55 | 78.29 | 83.35 | 84.65 | 78.3 | 86.82 | 84.96 |
| Nira(ours) | **86.08** | 1.53 | **86.17** | **85.42** | 1.82 | **85.51** | 84.48 | 1.99 | 84.61 | **84.67** | **2.1** | 84.79 |
| Nira Adapt(ours) | 86.08 | 1.53 | 86.17 | 85.26 | **1.66** | 85.39 | **84.59** | 1.84 | 84.71 | 84.33 | 2.67 | 84.48 |

Table 4: Acc, Atk Rate and Main Acc of defense algorithms on CIFAR-10, FMNIST and SVHN with different poison ratios.

| Poison Ratio | Defense | CIFAR-10 | | | FMNIST | | | SVHN | | |
|---|---|---|---|---|---|---|---|---|---|---|
| | | Acc | Atk Rate | Main Acc | Acc | Atk Rate | Main Acc | Acc | Atk Rate | Main Acc |
| 1% | FedAvg | 81.12 | 2.41 | 81.21 | 90.5 | 8.57 | 91.21 | 87.66 | 2.92 | 87.79 |
| | RFA | 84.62 | 1.55 | 84.72 | 91.49 | 1.9 | 91.59 | 89.25 | 1.23 | 89.32 |
| | Krum | 49.58 | 5.5 | 49.41 | 84.12 | 5.3 | 84.23 | 83.51 | 1.26 | 83.55 |
| | MultiKrum | 80.5 | 3.24 | 80.67 | 91.23 | 1.99 | 91.36 | 86.47 | 1.37 | 86.53 |
| | Coomed | 83.76 | 1.42 | 83.73 | **92** | 1.77 | **92.17** | 89.16 | 1.28 | 89.23 |
| | Normclip | 85.29 | 1.53 | 85.26 | 91.67 | 1.6 | 91.83 | 87.74 | 1.46 | 87.82 |
| | Nira(ours) | **85.59** | **1.2** | **85.71** | 91.36 | **1.6** | 91.43 | 90.45 | 0.99 | 90.52 |
| | Nira Adapt(ours) | 85.59 | 1.2 | 85.71 | 91.16 | 1.6 | 91.24 | **90.71** | **0.99** | **90.78** |
| 5% | FedAvg | 76.79 | 87.63 | 83.4 | 83.71 | 99.78 | **92.09** | 81.41 | 67.14 | 86.66 |
| | RFA | 79.38 | 76.31 | 85.2 | 87.41 | 22.74 | 89.2 | 88.99 | 5.36 | 89.36 |
| | Krum | 49.58 | 5.5 | 49.41 | 84.12 | 5.3 | 84.23 | 83.51 | 1.26 | 83.55 |
| | MultiKrum | 80.5 | 3.24 | 80.67 | 91.23 | 1.99 | 91.36 | 86.47 | 1.37 | 86.53 |
| | Coomed | 76.75 | 58.44 | 80.9 | 88.93 | 5.83 | 89.36 | 89.04 | 3.09 | 89.25 |
| | Normclip | 79.41 | 78.17 | **85.41** | 88.94 | 8.55 | 89.58 | 87.58 | 2.19 | 87.7 |
| | Nira(ours) | **85.2** | **2.3** | 85.31 | 91.46 | **1.44** | 91.55 | 90.72 | 1 | 90.75 |
| | Nira Adapt(ours) | 84.28 | 2.41 | 84.4 | 90.22 | 1.57 | 90.33 | **91.12** | **0.89** | **91.19** |
| 10% | FedAvg | 76.62 | 92.21 | 83.61 | 83.46 | 99.75 | **91.82** | 79.6 | 89.29 | 86.72 |
| | RFA | 78.44 | 84.71 | 84.95 | 85.4 | 28.68 | 87.63 | 87.34 | 10.51 | 88.05 |
| | Krum | 49.58 | 5.5 | 49.41 | 84.12 | 5.3 | 84.23 | 83.51 | 1.26 | 83.55 |
| | MultiKrum | 80.5 | 3.24 | 80.67 | 91.23 | 1.99 | 91.36 | 86.47 | 1.37 | 86.53 |
| | Coomed | 77.51 | 81.64 | 83.67 | 87.42 | 11.29 | 88.27 | 87.96 | 13.88 | 88.93 |
| | Normclip | 79.04 | 86.05 | **85.7** | 86.83 | 12.93 | 87.77 | 87.3 | 5.74 | 87.69 |
| | Nira(ours) | **85.38** | **2.06** | 85.52 | 91.32 | 1.57 | 91.37 | 90.6 | **0.82** | 90.67 |
| | Nira Adapt(ours) | 85.18 | 2.96 | 85.32 | 91.41 | **1.46** | 91.47 | **90.63** | 1.01 | **90.72** |
| 20% | FedAvg | 73.65 | 91.92 | 80.34 | 83.19 | 99.78 | **91.53** | 78.39 | 95.28 | 85.95 |
| | RFA | 76.47 | 87.71 | 83.08 | 80.03 | 34.62 | 82.38 | 84.08 | 35.51 | 86.73 |
| | Krum | 49.58 | 5.5 | 49.41 | 84.12 | 5.3 | 84.23 | 83.51 | 1.26 | 83.55 |
| | MultiKrum | 80.5 | 3.24 | 80.67 | 91.23 | 1.99 | 91.36 | 86.47 | 1.37 | 86.53 |
| | Coomed | 77.03 | 86.31 | 83.54 | 85.88 | 21.77 | 87.49 | 85.84 | 17.67 | 87.08 |
| | Normclip | 78.62 | 88.81 | 85.49 | 84.91 | 18.92 | 86.32 | 84.42 | 22.04 | 86.01 |
| | Nira(ours) | 85.53 | **2.01** | 85.71 | 90.91 | **1.64** | 90.95 | **91** | **0.76** | **91.05** |
| | Nira Adapt(ours) | **85.78** | 2.34 | **85.85** | **91.38** | 1.71 | 91.43 | 90.55 | 0.82 | 90.58 |

Normclip is sometimes slightly higher than Nira, its Atk Rate is much higher in comparison. This is mainly because Normclip aggregates all clipped local models, which helps with model convergence but does not completely eliminate the negative impact caused by attackers. On the other hand, Nira directly filters out malicious models by leveraging the surrogate dataset and does not select them in the aggregation process. Although this leads to a decrease in the number of clients participating in server aggregation, making it more challenging to converge, Nira still achieves a significantly lower Atk Rate than Normclip while maintaining comparable Acc and Main Acc.

## E.2 EXPERIMENTS WITH FEDPROX

FedProx (Li et al., 2020) is one of the more common training methods than FedAvg when extreme heterogeneity exists in the client data. Therefore, we conduct experiments with FedProx on CIFAR-10 in the Non-IID setting. We set the Non-IID degree control parameter $\alpha = 1$ and the poison ratio is 5%. The experimental results are shown in Table 3. It can be seen that Nira is easy to integrate with FedProx and performs well against backdoor attacks.

Table 5: Results of Nira with two filtering steps on CIFAR-10, FMNIST and SVHN datasets.

| Atk Num | Defense | CIFAR-10 | | | FMNIST | | | SVHN | | |
|---|---|---|---|---|---|---|---|---|---|---|
| | | Acc | Atk Rate | Main Acc | Acc | Atk Rate | Main Acc | Acc | Atk Rate | Main Acc |
| 0 | Nira | **86.29** | **1.33** | **86.28** | **91.2** | **1.38** | **91.26** | 90.32 | 0.82 | 90.39 |
| | Nira Acc | 86.21 | 1.35 | 86.26 | 91.03 | 1.38 | 91.15 | **90.34** | 1.04 | **90.78** |
| | Nira Feat | 86.05 | 1.46 | 86.07 | 90.48 | 2.01 | 90.57 | 82.31 | 1.41 | 82.41 |
| 4 | Nira | 85.2 | **2.3** | 85.31 | 91.46 | 1.44 | 91.55 | **90.72** | **1** | **90.75** |
| | Nira Acc | **85.22** | 2.34 | 85.59 | **91.75** | **1.42** | **91.82** | 89.28 | 1.45 | 89.62 |
| | Nira Feat | 79.09 | 85 | **85.65** | 84.05 | 40.28 | 87.16 | 85.26 | 31.64 | 87.6 |
| 8 | Nira | **85.25** | **2.49** | **85.42** | **90.79** | **1.82** | **90.89** | **90.68** | 0.88 | **90.72** |
| | Nira Acc | 85.08 | 2.69 | 85.15 | 90.59 | 1.93 | 90.68 | 88.98 | 4.1 | 89.44 |
| | Nira Feat | 78.06 | 89.75 | 84.97 | 83.7 | 50.46 | 87.76 | 85.03 | 12.86 | 85.92 |
| 12 | Nira | 84.68 | **2.87** | 84.9 | 87.29 | **3.07** | 87.38 | 89.48 | **1.44** | 89.52 |
| | Nira Acc | **84.8** | 2.93 | 84.99 | **88.14** | 9.87 | **88.82** | **89.52** | 9.56 | **90.26** |
| | Nira Feat | 78.89 | 90.63 | **85.96** | 81.18 | 68.28 | 86.69 | 87.02 | 25.83 | 88.98 |

Table 6: Performance of defense algorithms on CIFAR-10, FMNIST and SVHN when defending against varying attackers in the IID setting.

| Atk Num | Defense | CIFAR-10 | | | FMNIST | | | SVHN | | |
|---|---|---|---|---|---|---|---|---|---|---|
| | | Acc | Atk Rate | Main Acc | Acc | Atk Rate | Main Acc | Acc | Atk Rate | Main Acc |
| 0 | FedAvg | **79** | 2.32 | **78.89** | 91.99 | 1.86 | **92.09** | **87.63** | 1.54 | **87.99** |
| | RFA | 78.65 | 2.34 | 78.55 | 91.51 | 1.97 | 91.61 | 87.49 | 1.48 | 87.81 |
| | Krum | 61.73 | 7.39 | 61.62 | 87.2 | 2.17 | 87.19 | 79.89 | 3.05 | 80.27 |
| | MultiKrum | 76.49 | 2.21 | 76.37 | 90.7 | 1.99 | 90.78 | 85.62 | 2.17 | 85.94 |
| | Coomed | 78.35 | 2.14 | 78.2 | **92.01** | 1.79 | 92.09 | 87.56 | 1.55 | 87.76 |
| | Normclip | 77 | 2.47 | 76.87 | 91.65 | 1.84 | 91.7 | 82.6 | 2.52 | 82.78 |
| | Nira(ours) | 78.98 | **1.46** | 78.8 | 90.97 | **1.55** | 91 | 86.44 | **1.01** | 86.5 |
| | Nira Adapt(ours) | 78.98 | 1.46 | 78.8 | 90.97 | 1.55 | 91 | 86.44 | 1.01 | 86.5 |
| 4 | FedAvg | 74.32 | 62.6 | 78.46 | 83.81 | 95.65 | **91.84** | 80.08 | 97.08 | **87.99** |
| | RFA | 74.86 | 60.96 | **78.89** | 88.71 | 19.6 | 90.32 | 80.21 | 96.68 | 88.1 |
| | Krum | 58.42 | 6.38 | 58.37 | 87.52 | 2.63 | 87.47 | 81.11 | 2.66 | 81.44 |
| | MultiKrum | 76.13 | 2.55 | 76.11 | 90.77 | 2.25 | 90.89 | 85.63 | 1.8 | 85.98 |
| | Coomed | 76.82 | 31.82 | 78.63 | 89.44 | 19.03 | 90.82 | 79.37 | 73.9 | 85.11 |
| | Normclip | 73.15 | 59.03 | 76.86 | 89.89 | 5.3 | 90.27 | 82.63 | 3 | 82.84 |
| | Nira(ours) | 78.57 | **1.75** | 78.37 | **91.33** | **1.64** | 91.41 | **85.91** | **1.06** | 86.05 |
| | Nira Adapt(ours) | **78.93** | 1.97 | 78.77 | 90.64 | 2.13 | 90.74 | 84.85 | 1.3 | 84.98 |
| 8 | FedAvg | 73.38 | 77.93 | 78.76 | 83.58 | 98.24 | **91.83** | 79.7 | 98.85 | **87.73** |
| | RFA | 73.52 | 76.79 | **78.83** | 86.67 | 29.45 | 89.08 | 79.67 | 98.88 | 87.71 |
| | Krum | 59.75 | 6.09 | 59.73 | 85.88 | 2.74 | 86.03 | 80 | 4.25 | 81.44 |
| | MultiKrum | 76.12 | 4.16 | 76.1 | 90.91 | 2.54 | 91.04 | 85.01 | 5.43 | 85.67 |
| | Coomed | 73.29 | 73.28 | 78.28 | 86.76 | 31.88 | 89.3 | 79.77 | 98.43 | 87.77 |
| | Normclip | 71.93 | 73.48 | 76.78 | 87.81 | 19.36 | 89.3 | 82.51 | 3.23 | 82.71 |
| | Nira(ours) | **78.32** | 2.19 | 78.21 | 90.68 | **2.02** | 90.87 | 84.48 | 1.18 | 84.64 |
| | Nira Adapt(ours) | 78.01 | 2.3 | 77.89 | **91.1** | 2.52 | 91.25 | **85.47** | **1.15** | 85.64 |
| 12 | FedAvg | 73.21 | 81.77 | **78.94** | 83.63 | 99.07 | **91.95** | 79.82 | 99.55 | **87.94** |
| | RFA | 72.95 | 81.2 | 78.63 | 84.16 | 59.6 | 89.01 | 79.57 | 99.49 | 87.65 |
| | Krum | 60.76 | 7.41 | 60.73 | 86.4 | 4.16 | 86.48 | 76.34 | 64.06 | 81.04 |
| | MultiKrum | 76.14 | 5.29 | 76.25 | 89.88 | 5.41 | 90.22 | 77.72 | 96.6 | 85.35 |
| | Coomed | 73.24 | 80.08 | 78.83 | 84.42 | 56.46 | 89.03 | 79.59 | 99.33 | 87.66 |
| | Normclip | 71.49 | 79.38 | 76.87 | 85.03 | 52.32 | 89.27 | 82.28 | 3.92 | 82.52 |
| | Nira(ours) | **77.55** | **2.41** | 77.38 | **90.06** | 3.13 | 90.19 | **85.19** | 1.61 | 85.35 |
| | Nira Adapt(ours) | 76.56 | 2.63 | 76.36 | 89.3 | 3.39 | 89.42 | 84.59 | **1.35** | 84.67 |

### E.3 PERFORMANCE IN THE IID SETTING

To further demonstrate the applicability of Nira, we compared the performance of defense methods on 3 datasets in the IID setting. We set the Non-IID degree control parameter $\alpha = 100$ to simulate the IID setting. The experimental results are shown in Table 6 and 7. It can be seen that Nira can still effectively defend against backdoor attacks in the IID setting and preserve high accuracy simultaneously.

Table 7: Performance of defense algorithms on CIFAR-10, FMNIST and SVHN with different poison ratios in the IID setting.

| Poison Ratio | Defense | CIFAR-10 | | | FMNIST | | | SVHN | | |
|---|---|---|---|---|---|---|---|---|---|---|
| | | Acc | Atk Rate | Main Acc | Acc | Atk Rate | Main Acc | Acc | Atk Rate | Main Acc |
| 1% | FedAvg | **79** | 2.32 | **78.89** | 91.99 | 1.86 | **92.09** | **87.64** | 1.54 | **87.99** |
| | RFA | 78.65 | 60.96 | 78.89 | 91.51 | 1.97 | 91.61 | 87.49 | 1.48 | 87.81 |
| | Krum | 61.73 | 7.39 | 61.62 | 87.2 | 2.17 | 87.19 | 79.89 | 3.05 | 80.27 |
| | MultiKrum | 76.49 | 2.21 | 76.37 | 90.7 | 1.99 | 90.78 | 85.62 | 2.17 | 85.94 |
| | Coomed | 78.35 | 2.14 | 78.2 | **92.01** | 1.79 | 92.09 | 87.56 | 1.55 | 87.76 |
| | Normclip | 77 | 2.47 | 76.87 | 91.65 | 1.84 | 91.7 | 82.6 | 2.52 | 82.78 |
| | Nira(ours) | 78.97 | **1.58** | 78.79 | 90.97 | **1.55** | 91 | 85.64 | **1.04** | 85.93 |
| | Nira Adapt(ours) | 78.97 | 1.58 | 78.79 | 90.97 | 1.55 | 91 | 85.29 | 1.11 | 85.55 |
| 5% | FedAvg | 74.32 | 62.6 | 78.46 | 83.81 | 95.65 | **91.84** | 80.08 | 97.08 | 87.99 |
| | RFA | 74.86 | 60.96 | **78.89** | 88.71 | 19.6 | 90.32 | 80.21 | 96.68 | **88.1** |
| | Krum | 58.42 | 6.38 | 58.37 | 87.52 | 2.63 | 87.47 | 81.11 | 2.66 | 81.44 |
| | MultiKrum | 76.13 | 2.55 | 76.11 | 90.77 | 2.25 | 90.89 | 85.63 | 1.8 | 85.98 |
| | Coomed | 76.82 | 31.82 | 78.63 | 89.44 | 19.03 | 90.82 | 79.37 | 73.9 | 85.11 |
| | Normclip | 73.15 | 59.03 | 76.86 | 89.89 | 5.3 | 90.27 | 82.63 | 3 | 82.84 |
| | Nira(ours) | 78.57 | **1.75** | 78.37 | **91.33** | **1.64** | 91.41 | **85.91** | **1.06** | 86.05 |
| | Nira Adapt(ours) | **78.93** | 1.97 | 78.77 | 90.64 | 2.13 | 90.74 | 84.85 | 1.3 | 84.98 |
| 10% | FedAvg | 74.1 | 72.82 | **79.1** | 83.65 | 97.98 | **91.89** | 79.89 | 99.42 | **88** |
| | RFA | 73.81 | 71.4 | 78.71 | 86.96 | 37.93 | 90.05 | 79.68 | 99.51 | 87.78 |
| | Krum | 56.71 | 7.01 | 56.68 | 88.64 | 2.19 | 88.61 | 81.11 | 2.66 | 81.44 |
| | MultiKrum | 76.97 | 3 | 76.87 | **90.94** | 2.14 | 91 | 85.67 | 1.7 | 86.02 |
| | Coomed | 74.59 | 62.85 | 78.75 | 87.97 | 29.27 | 90.33 | 79.9 | 98.18 | 87.88 |
| | Normclip | 72.17 | 70.72 | 76.86 | 86.96 | 37.01 | 89.95 | 82.33 | 3.46 | 82.53 |
| | Nira(ours) | **78.47** | **1.8** | 78.24 | 90.57 | **1.73** | 90.63 | **85.75** | **1.2** | 85.91 |
| | Nira Adapt(ours) | 78.15 | 2.43 | 78.08 | 90.71 | 2.17 | 90.81 | 84.43 | 1.53 | 84.58 |
| 20% | FedAvg | 73.33 | 79.03 | **78.84** | 83.58 | 99.14 | **91.91** | 79.75 | 100 | **87.9** |
| | RFA | 72.85 | 78.44 | 78.27 | 85.25 | 49.64 | 89.3 | 79.66 | 100 | 87.81 |
| | Krum | 57.95 | 7.65 | 57.95 | 88.64 | 2.19 | 88.61 | 81.11 | 2.66 | 81.44 |
| | MultiKrum | 76.42 | 3.78 | 76.44 | 90.76 | 2.21 | 90.88 | 85.58 | 1.98 | 85.99 |
| | Coomed | 73.48 | 74.91 | 78.61 | 85.55 | 56.05 | 90.14 | 79.67 | 99.84 | 87.88 |
| | Normclip | 71.54 | 77.85 | 76.76 | 85.1 | 52.67 | 89.43 | 82.14 | 4.86 | 82.4 |
| | Nira(ours) | 78.59 | **1.83** | 78.43 | **90.95** | **1.71** | 91.02 | **85.91** | **1.16** | 86.1 |
| | Nira Adapt(ours) | **78.64** | 2.29 | 78.61 | 90.88 | 2.39 | 90.99 | 84.55 | 1.35 | 84.63 |

Table 8: Performance of Nira with interval-based filtering strategy on CIFAR-10, FMNIST and SVHN datasets.

| Atk Num | Defense | CIFAR-10 | | | FMNIST | | | SVHN | | |
|---|---|---|---|---|---|---|---|---|---|---|
| | | Acc | Atk Rate | Main Acc | Acc | Atk Rate | Main Acc | Acc | Atk Rate | Main Acc |
| 0 | FedAvg | 84.72 | 3.26 | 84.87 | **91.51** | 1.77 | **91.64** | 89.2 | 1.12 | 89.28 |
| | Nira | 86.29 | **1.33** | 86.28 | 91.2 | 1.38 | 91.26 | 90.32 | **0.82** | 90.39 |
| | Nira Interval | **86.43** | 1.41 | **86.57** | 91.3 | **1.36** | 91.38 | **90.51** | 0.86 | **90.58** |
| 4 | FedAvg | 76.79 | 87.63 | 83.4 | 83.71 | 99.78 | **92.09** | 81.41 | 67.14 | 86.66 |
| | Nira | 85.2 | **2.3** | 85.31 | **91.46** | 1.44 | 91.55 | 90.72 | **1** | 90.75 |
| | Nira Interval | **85.46** | 2.47 | **85.82** | 91.38 | **1.4** | 91.45 | **90.83** | 1.18 | **90.91** |
| 8 | FedAvg | 77.86 | 88.9 | 84.67 | 83.27 | 99.67 | **91.61** | 80.79 | 66.21 | 85.9 |
| | Nira | 85.25 | 2.49 | 85.42 | **90.79** | **1.82** | 90.89 | **90.68** | **0.88** | **90.72** |
| | Nira Interval | **85.36** | **2.35** | **85.57** | 89.85 | 1.83 | 90.02 | 90.32 | 1.28 | 90.42 |
| 12 | FedAvg | 77.8 | 89.64 | 84.67 | 83.34 | 99.64 | **91.68** | 81.58 | 75.29 | 87.61 |
| | Nira | 84.68 | 2.87 | 84.9 | 87.29 | 3.07 | 87.38 | 89.48 | 1.44 | 89.52 |
| | Nira Interval | **84.97** | **2.42** | **85.05** | **88.12** | **2.96** | 88.28 | **89.92** | **1.33** | **90.17** |

## E.4 Impacts of Two Filtering Steps

To investigate the individual effects of the two filtering steps in Nira, we conduct experiments on 3 datasets with varying attackers. We compare the defensive capabilities of Nira with both filtering steps (Nira), Nira with only the accuracy test (Nira Acc), and Nira with only the feature distribution test (Nira Feat). The experimental results are shown in Table 5.

We can see that in most cases, Nira Acc demonstrates commendable performance, exhibiting a defense capability comparable to that of Nira. However, in certain cases, such as when the FMNIST dataset or the SVHN dataset involves 12 attackers, the Atk Rate of Nira Acc is much higher than that of Nira. This discrepancy suggests that while the accuracy test effectively identifies the majority of

attackers, it still fails to detect a small fraction, thereby causing damage to the global model. It is worth noting that despite the unsatisfactory performance of Nira Feat, the integration of both testing steps in Nira enhances its defense capability. This implies that the feature distribution test serves as a valuable complement to the accuracy test, aiding the server in effectively identifying attackers that may have been missed during the accuracy assessment.

### E.5 DIFFERENT FILTERING STRATEGY

Nira utilizes two filtering steps based on the threshold $\sigma_1$ and $\sigma_2$ to filter out attackers. We assume that the server can identify a small number of benign clients and simulate the training process. Then we can have an efficient method for selecting thresholds: first, train a few rounds on a small number of identified benign clients, referring to the accuracy of the model on surrogate data and the similarity of features during the training process to assist in selecting $\sigma_1$ and $\sigma_2$. Then the selected $\sigma_1$ and $\sigma_2$ are used for the training of all clients. The thresholds at the beginning of training are relatively crucial, because if attackers are successfully filtered out in the early stages, the accuracy of attackers on surrogate data will have a significant gap compared to the accuracy of benign clients. This allows for a wider range of threshold selection in subsequent rounds.

Nevertheless, when the proportion of attackers is significant, there may be significant fluctuations in the appropriate threshold values during each training round, making it more difficult to accurately filter out attackers. To address this issue, we propose an interval-based filtering strategy. The core idea is that benign clients and malicious clients behave differently on the surrogate dataset. Benign models exhibit relatively higher accuracy on noise data and have smaller intervals between them, while malicious models show relatively lower accuracy on noise data and have larger intervals compared to benign models. Therefore, we can set an interval $\epsilon_1$ to filter out malicious models. Specifically, for the local models $\{L_1, \ldots, L_n\}$, we assume that $[L_{[1]}, \ldots, L_{[n]}]$ is an ordering of model accuracy from high to low. Then we can get a set $S$ of benign models based on the interval $\epsilon_1$:

$$S = \max_k \{L_{[1]}, \ldots, L_{[k]} | acc(L_{[i]}) - acc(L_{[i+1]}) \leq \epsilon_1, \forall 1 \leq i < k\}. \tag{15}$$

Similarly, we can also set an interval $\epsilon_2$ for feature distance. Consequently, we can change the threshold-based filtering strategy to interval-based. The experimental results are shown in Table 8. It can be seen that Nira using the interval-based filtering strategy can also perform well in defending against backdoor attackers.

### E.6 SURROGATE DATASET GENERATION

To further investigate the effect of different surrogate datasets, we employ two additional generated datasets to replace the original surrogate dataset produced by StyleGAN. These two datasets include one generated by a simple CNN and another generated by upsampling pure Gaussian noise. In our data generation methods, we sample noise from various Gaussian distributions, each with the same mean but different standard deviations, to generate noise with diverse latent styles that correspond to distinct classes. Given that datasets CIFAR-10, FMNIST and SVHN each consist of 10 classes, the surrogate dataset also comprises 10 classes. The size of the surrogate dataset is 2000 in our experiments, which is a small proportion of the utilized datasets, i.e., $3.33\%$ for CIFAR-10, $2.86\%$ for FMNIST, and $0.33\%$ for SVHN. We show the generated surrogate datasets as Figure 5, 6 and 7. We also conduct ablations on the sensitivity of the size of the surrogate dataset and report results in Table 17.

For the dataset generated by the simple CNN, we first sample 64-dimensional noises. These noises are then fed into a CNN composed of 4 transpose convolutional layers and 3 convolutional layers. The CNN model processes the input and produces noise data of size $32 \times 32$. We employ 10 CNNs with distinct initial weights to generate the noise data that have enough diversity as the dataset of 10 classes.

For the dataset generated by upsampling the pure Gaussian noise, noise points are initially sampled to form an image of size $8 \times 8$. Subsequently, upsampling is employed to transform the image into a larger size of $32 \times 32$. This upsampling process enables the generation of noise images with some low-level features, thereby enabling the model to learn basic feature distributions from them.

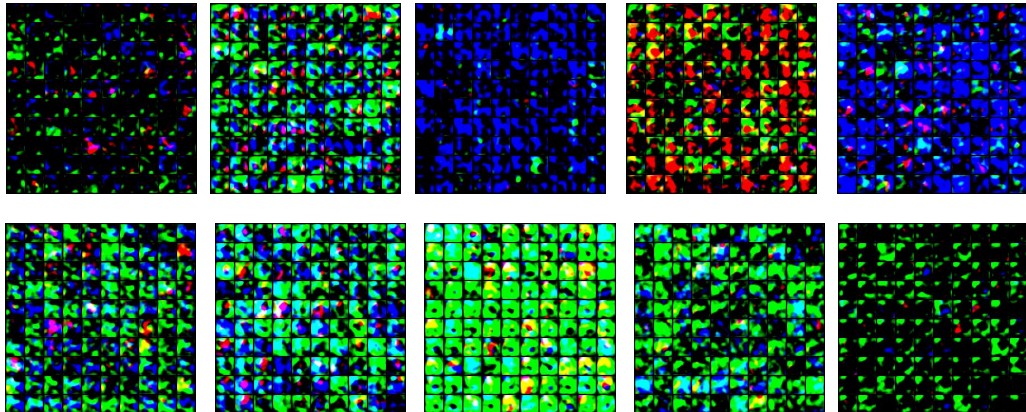

Figure 5: Surrogate dataset generated by the StyleGAN.

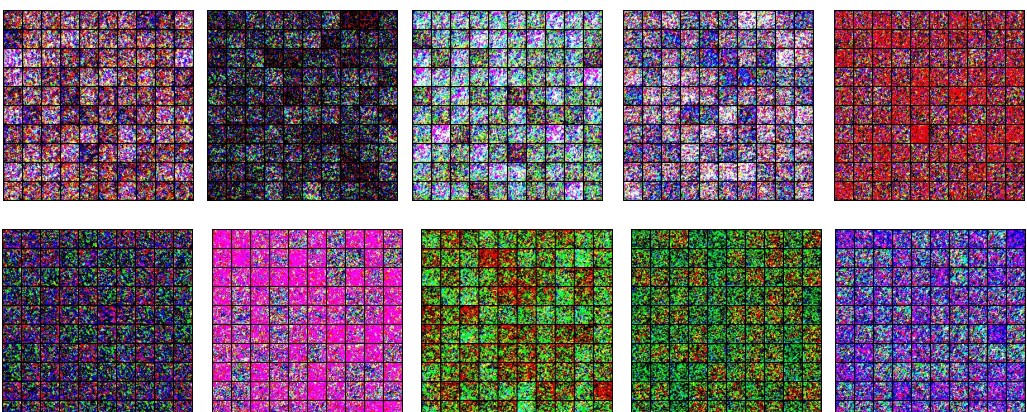

Figure 6: Surrogate dataset generated by the simple CNN.

As shown in Table 9, surrogate datasets generated by these two methods can also provide powerful defense capabilities to the server, which further demonstrates the applicability and relevance of Nira.

### E.7 DIFFERENT GLOBAL MODEL

We investigate the sensitivity of Nira to various shared global models. In particular, we conduct experiments on CIFAR-10 to compare the performance of Nira with different defense algorithms on a range of models, including ResNet-10, ResNet-34 (He et al., 2016), VGG-9 and VGG-19 (Simonyan & Zisserman, 2014). The results presented in Table 10 demonstrate the effectiveness of Nira across models of varying capacities.

### E.8 DIFFERENT HYPERPARAMETER $\lambda$

We adjust the align weight $\lambda$ in the Nira objective $F_k^{Nira}$ from 0.1 to 5 on CIFAR-10 to examine the sensitivity of Nira to $\lambda$. The results in Table 11 demonstrate that Nira is not sensitive to the align weight $\lambda$, and it can achieve good performance within a wide range of $\lambda$.

### E.9 MORE ADAPTIVE ATTACK SCENARIOS

In Table 1, 2, 3, 4, 10, 6 and 7, we show the results of Nira against the adaptive attack scenario: The attackers divide their poisoned dataset into poisoned and benign parts, and only align the surrogate samples with the data within the benign parts. Empirical results show that Nira still performs rela-

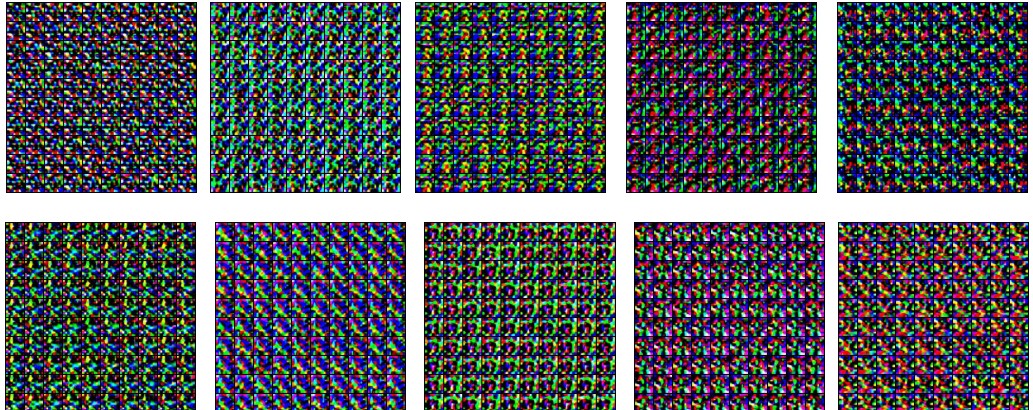

Figure 7: Surrogate dataset generated by upsampling the pure Gaussian noise.

Table 9: Results of Nira with different generated noise datasets on CIFAR-10, FMNIST and SVHN.

| Atk Num | Defense | CIFAR-10 | | | FMNIST | | | SVHN | | |
|---|---|---|---|---|---|---|---|---|---|---|
| | | Acc | Atk Rate | Main Acc | Acc | Atk Rate | Main Acc | Acc | Atk Rate | Main Acc |
| 0 | FedAvg | 84.72 | 3.26 | 84.87 | **91.51** | 1.77 | **91.64** | 89.2 | 1.12 | 89.28 |
| | Nira | **86.29** | **1.33** | **86.28** | 91.2 | **1.38** | 91.26 | **90.32** | **0.82** | **90.39** |
| | Nira Gaus | 84.61 | 1.66 | 84.71 | 91.09 | 1.71 | 91.21 | 90.09 | 0.83 | 90.15 |
| | Nira CNN | 84.01 | 1.51 | 84.02 | 90.78 | 1.71 | 90.91 | 89.97 | 1.25 | 90.05 |
| 4 | FedAvg | 76.79 | 87.63 | 83.4 | 83.71 | 99.78 | 92.09 | 81.41 | 67.14 | 86.66 |
| | Nira | **85.2** | **2.3** | **85.31** | **91.46** | **1.44** | **91.55** | 90.72 | 1 | 90.75 |
| | Nira Gaus | 81.67 | 2.43 | 81.82 | 90.49 | 1.9 | 90.62 | **90.88** | **0.85** | **90.95** |
| | Nira CNN | 81.82 | 2.87 | 81.88 | 90.31 | 2.03 | 90.37 | 89.18 | 1.4 | 89.27 |
| 8 | FedAvg | 77.86 | 88.9 | 84.67 | 83.27 | 99.67 | **91.61** | 80.79 | 66.21 | 85.9 |
| | Nira | **85.25** | 2.49 | **85.42** | 90.79 | 1.82 | 90.89 | **90.68** | **0.88** | **90.72** |
| | Nira Gaus | 83.16 | **1.46** | 83.22 | **90.84** | 1.88 | 90.97 | 89.74 | 3.18 | 89.9 |
| | Nira CNN | 80.47 | 1.6 | 80.49 | 90.37 | **1.55** | 90.45 | 89.68 | 3.36 | 89.87 |
| 12 | FedAvg | 77.8 | 89.64 | 84.67 | 83.34 | 99.64 | **91.68** | 81.58 | 75.29 | 87.61 |
| | Nira | **84.68** | 2.87 | **84.9** | **87.29** | 3.07 | 87.38 | **89.48** | **1.44** | **89.52** |
| | Nira Gaus | 84.41 | **2.23** | 84.49 | 85.54 | 3.37 | 85.64 | 87.19 | 3.8 | 87.41 |
| | Nira CNN | 80.97 | 2.67 | 80.96 | 85.99 | 6.31 | 86.37 | 85.45 | 4.56 | 85.71 |

tively well against such an adaptive attack. We infer that Nira is still effective under this adaptive attack for two main reasons: 1) The scale-up operation on the model weights in the model replacement attack influences the alignment of the distribution. 2) The attacker does not align the features of the poisoned samples with the surrogate data, leading to a greater influence of the cross-entropy term in Eq. 3. The role of poisoned samples in the cross-entropy term influences the test performances.

We also consider the performance of Nira under another adaptive attack scenario: the attacker changes the alignment parameter $\lambda$ used to a different one not used by the benign clients. The experimental results are shown in Table 12.

### E.10 CURVES DURING THE TRAINING

We present the training phase curves of Attack Rate (Atk Rate) and Main Accuracy (Main Acc) on 3 datasets with the partition parameter $\alpha = 1$. These curves are shown in Figure 10, 11 and 12. In the figures, the grey line represents the actual values, while the solid line corresponds to the smoothed curve derived from the original data, providing enhanced visibility. The presented curves illustrate the effectiveness of Nira in accelerating the convergence of the global model, improving model accuracy, and effectively mitigating backdoor attacks. It is worth noting that the curves have several downward dips where the accuracy goes down for a few rounds before recovering. Especially, it happens even without any attackers. Through scrutinizing the training process, we discovered that during several rounds of decreasing curves, similar client subsets were selected for

Table 10: Performance of different defense algorithms on different models on CIFAR-10.

| Defense | ResNet-10 | | | ResNet-34 | | | VGG-9 | | | VGG-19 | | |
|---|---|---|---|---|---|---|---|---|---|---|---|---|
| | Acc | Atk Rate | Main Acc | Acc | Atk Rate | Main Acc | Acc | Atk Rate | Main Acc | Acc | Atk Rate | Main Acc |
| FedAvg | 72.07 | 85.32 | 78.09 | 73.36 | 88.13 | 79.69 | 45 | 8.57 | 44.87 | 45.33 | 9.42 | 45.28 |
| RFA | 76.68 | 73.75 | **82** | 76.88 | 71.42 | 82.09 | 45.53 | 6.09 | 45.22 | 41.53 | 7.19 | 41.44 |
| Krum | 53.44 | 6.33 | 53.36 | 43.42 | 7.63 | 43.24 | 26 | 29.21 | 26.39 | 18.44 | 45.17 | 19.01 |
| MultiKrum | 75.04 | 2.47 | 75 | 80.46 | 3.33 | 80.59 | 43.04 | 6.95 | 42.82 | 43.7 | 10.13 | 43.62 |
| Coomed | 76.18 | 62.25 | 80.49 | 79.11 | 65.28 | 84 | 32.2 | 5.98 | 32.1 | 23.37 | 3.2 | 23.34 |
| Normclip | 75.13 | 66 | 79.7 | 78.39 | 77.12 | 84.19 | 45.96 | 4.64 | 45.57 | 42.55 | 7.08 | 42.46 |
| Nira(ours) | **80.28** | **1.73** | 80.46 | **86.53** | **1.71** | **86.91** | **56.63** | **1.58** | **56.95** | **55.1** | 2.52 | **55.91** |
| Nira Adapt(ours) | 79.54 | 2.4 | 79.77 | 85.48 | 2.02 | 85.68 | 55.84 | 1.72 | 56.01 | 53.74 | 3.94 | 54.76 |

Table 11: Performance of Nira on CIFAR-10 for varying align parameter $\lambda$.

| $\lambda$ | Atk Num $= 0$ | | | Atk Num $= 4$ | | | Atk Num $= 8$ | | | Atk Num $= 12$ | | |
|---|---|---|---|---|---|---|---|---|---|---|---|---|
| | Acc | Atk Rate | Main Acc | Acc | Atk Rate | Main Acc | Acc | Atk Rate | Main Acc | Acc | Atk Rate | Main Acc |
| $\lambda = 0.1$ | 83.94 | 2.76 | 84.03 | 83.11 | 2.93 | 83.23 | 82.84 | 2.01 | 82.94 | 82.22 | 2.85 | 82.34 |
| $\lambda = 0.2$ | 84.45 | 2.25 | 84.56 | 83.66 | 3.09 | 83.82 | 83.33 | 2.23 | 83.44 | 83.01 | 3.24 | 83.16 |
| $\lambda = 0.5$ | 85.96 | 1.86 | 85.98 | 84.68 | 2.57 | 84.8 | 84.24 | 2.27 | 84.39 | 84.08 | 3.05 | 84.25 |
| $\lambda = 1$ | 86.29 | **1.33** | 86.28 | 85.2 | 2.3 | 85.31 | 85.25 | 2.49 | 85.42 | 84.68 | 2.87 | 84.9 |
| $\lambda = 2$ | 88.12 | 1.67 | 88.32 | 85.71 | **1.61** | 85.74 | 85.51 | 2.37 | 85.66 | 85.37 | 2.24 | 85.46 |
| $\lambda = 5$ | **88.58** | 1.82 | **88.73** | **85.87** | 2.12 | **86.06** | **85.87** | **1.46** | **85.98** | **86.22** | **1.49** | **86.37** |

aggregation. Considering that we conduct experiments in the non-IID setting, we infer the reason for this phenomenon is that aggregating similar client subsets continuously leads to overfitting of the model on the local data of these clients, resulting in a decrease in the overall accuracy. This phenomenon caused by randomly selecting client subsets will still exist even without any attackers.

We also present the Defense Rate curves obtained during the training phase on 3 datasets, with a maximum of 8 attackers, as illustrated in Figure 13. The results demonstrate that Nira successfully filters out all attackers across the three datasets when the number of attackers is set to 4. Furthermore, even when the number of attackers increases to 8, Nira continues to effectively filter out all attackers on CIFAR-10 and successfully filters out attackers most of the time on FMNIST and SVHN. It is worth noting that Nira exhibits minimal instances of missing attackers after 70 rounds, highlighting its ability to consistently enhance the accuracy of attacker detection during training and mitigate the impact of attackers on the final global model.

### E.11 MORE ABLATION EXPERIMENTS

To further demonstrate the effectiveness of Nira, we conduct more ablation experiments on CIFAR-10. We consider a new scenario: a total of 50 clients, all participating in aggregation each round. Table 13 and 14 show the performance of different defense algorithms under large attacker ratios and large poison ratios respectively, which further underscore the robust effectiveness of Nira in diverse settings. In Table 18, we conduct experiments without the scale-up operation. We can see that the proposed method still outperforms baselines in the absence of the scale-up operation.

Malicious attacks may inject backdoor triggers into the surrogate data. However, if an attacker attempts to mount a backdoor attack on the surrogate data, the model sent from the attacker will achieve poor performance on the surrogate data shared among clients and server. As a result, the server will filter out these malicious models by evaluating their performance on the surrogate data. Therefore, injecting backdoor triggers on the surrogate data will also be detected by the proposed method. To verify the point, we inject backdoor triggers into the surrogate dataset on a malicious client while other settings are the same as our original setting. The results are reported in Table 15. The results show that these malicious clients will not bypass the proposed method.

In Table 16, we conduct ablations on the sensitivity of b mentioned in Section 3.3. We can see that b has a limited impact on the performance. In Table 17, we investigate the effect of the size of the surrogate dataset on the performance. We can see that the size of the surrogate dataset has a limited impact on the performance. Even when the surrogate dataset size is only 500, the proposed method still demonstrates excellent performance.

Table 19 shows the results of only one attacker and no attackers under different poison ratios for both FedAvg and Nira. Figure 8 shows the accuracy of a benign client and a malicious client, where

Table 12: Performance of Nira under changing attack $\lambda$.

| Atk Num | Metrics | Atk $\lambda$ | | | | | | | |
|---------|---------|------|------|------|------|------|------|------|------|
| | | 0.01 | 0.1 | 0.5 | 1 | 2 | 5 | 10 | 100 |
| 4 | Acc | 85.23 | **85.26** | 85.09 | 85.2 | 84.96 | 85.21 | 85.15 | 79.75 |
| | Atk Rate | 2.54 | 2.21 | 2.37 | 2.3 | **2.14** | 2.6 | 2.41 | 2.3 |
| | Main Acc | 85.31 | 85.36 | 85.14 | 85.31 | 85.14 | 85.36 | **85.4** | 79.79 |
| 8 | Acc | 84.31 | 85.05 | 84.91 | **85.25** | 85 | 85.05 | 84.72 | 72.77 |
| | Atk Rate | **2.01** | 2.3 | 2.2 | 2.49 | 2.26 | 2.36 | 2.17 | 2.01 |
| | Main Acc | 84.46 | 85.21 | 85.01 | **85.42** | 85.15 | 85.18 | 84.9 | 72.62 |

Table 13: Performance of different defense algorithms under large attacker ratios.

| Defense | Atk Num = 0 | | | Atk Num = 10 | | | Atk Num = 20 | | | Atk Num = 30 | | |
|---------|-----|----------|----------|-----|----------|----------|-----|----------|----------|-----|----------|----------|
| | Acc | Atk Rate | Main Acc | Acc | Atk Rate | Main Acc | Acc | Atk Rate | Main Acc | Acc | Atk Rate | Main Acc |
| FedAvg | 89.35 | **0.67** | 89.5 | 79.46 | 97.23 | 87.17 | 81.61 | 94.82 | **89.3** | 81.99 | 96.92 | **89.94** |
| RFA | 87.51 | 1.16 | 87.56 | 80.83 | 81.25 | 87.24 | 80.4 | 86.75 | 87.23 | 79.67 | 91.88 | 86.91 |
| MultiKrum | 88.33 | 1.22 | 88.4 | 88.32 | 1.35 | 88.47 | 80.35 | 88.31 | 87.3 | 81.03 | 97.01 | 88.88 |
| Coomed | 87.91 | 0.96 | 87.91 | 81.41 | 89.1 | 88.57 | 80.94 | 91.51 | 88.27 | 80.64 | 95.46 | 88.3 |
| Normclip | 79.71 | 1 | 79.65 | 73.8 | 83.57 | 79.77 | 73.16 | 87.8 | 79.43 | 73.26 | 89.8 | 79.68 |
| Nira(ours) | **89.74** | 0.94 | **89.78** | **88.85** | 1.24 | **88.92** | **87.67** | 2.23 | 87.88 | **85.13** | 3.46 | 85.38 |

the benign data of both clients is fixed to be the same while introducing additional poisoned samples to the malicious client. We can see that the client introducing poisoned samples performs worse on noise samples, making it easier for the server to identify attackers. Figure 9 shows the alignment loss between poisoned data and surrogate data with or without the scale-up operation. We can see that the scale-up operation increases the alignment loss between poisoned data and surrogate data.

# F MORE VISUALIZATION RESULTS

In this section, we provide additional visualization results of the feature distributions for both the server and attacker. Figure 14 shows the features of the server in FedAvg. Figure 15, 16 and 17 show the features of the server and attacker on CIFAR-10, FMNIST, SVHN respectively. It's easy to observe that the feature distribution of the attacker is more scattered and less cohesive compared to the server, which leads to worse performance of the local model on the surrogate noise data and enables the server to identify attackers.

Table 14: Performance of different defense algorithms under large poison ratios.

| Defense | Poison Ratio = 20% | | | Poison Ratio = 40% | | | Poison Ratio = 60% | | | Poison Ratio = 80% | | |
|---|---|---|---|---|---|---|---|---|---|---|---|---|
| | Acc | Atk Rate | Main Acc | Acc | Atk Rate | Main Acc | Acc | Atk Rate | Main Acc | Acc | Atk Rate | Main Acc |
| FedAvg | 81.1 | 99.25 | **89.16** | 80.48 | 99.75 | **88.53** | 79.05 | 99.69 | **86.95** | 76.92 | 99.78 | 84.62 |
| RFA | 79.44 | 89.51 | 86.71 | 78.12 | 88.55 | 85.07 | 75.95 | 92.21 | 83.03 | 74.95 | 85.54 | 81.54 |
| MultiKrum | 80.36 | 99.47 | 88.37 | 79.9 | 99.75 | 87.9 | 77.88 | 99.69 | 85.67 | 74.92 | 99.78 | 82.42 |
| Coomed | 80.28 | 98.04 | 88.15 | 79.67 | 99.32 | 87.6 | 78.29 | 99.78 | 86.13 | 75.01 | 99.75 | 82.51 |
| Normclip | 71.92 | 92.34 | 78.45 | 69.27 | 93.13 | 75.62 | 66.38 | 93.55 | 72.48 | 61.66 | 93.79 | 67.31 |
| Nira(ours) | **84.78** | **4.34** | 85.09 | **84.9** | **4.58** | 85.16 | **85.02** | **3.92** | 85.23 | **84.8** | **4.4** | **85.09** |

Table 15: Results of Nira with poisoned surrogate data.

| Defense | Metric | Surrogate Data Poison Ratio | | | | | | |
|---|---|---|---|---|---|---|---|---|
| | | 0% | 5% | 10% | 20% | 40% | 60% | 80% |
| Nira | Acc | 87.67 | 87.59 | **88** | 87.78 | 87.41 | 87.54 | 87.02 |
| | Atk Rate | **2.23** | 2.23 | 2.41 | 2.65 | 3.09 | 2.47 | 3.15 |
| | Main Acc | 87.88 | 87.83 | **88.22** | 88.05 | 87.79 | 87.82 | 87.22 |

Table 16: Results of Nira with different noise sample number.

| Defense | Metric | Noise Sample Num $b$ | | | | |
|---|---|---|---|---|---|---|
| | | 64 | 128 | 256 | 512 | 1024 |
| Nira | Acc | 87.31 | 87.67 | **87.7** | 87.54 | 87.24 |
| | Atk Rate | 1.92 | 2.23 | 2.89 | **1.9** | 2.69 |
| | Main Acc | 87.42 | **87.88** | 87.83 | 87.69 | 87.49 |

Table 17: Results of Nira with different surrogate dataset size.

| Defense | Metric | Surrogate Dataset Size | | | | | | |
|---|---|---|---|---|---|---|---|---|
| | | 500 | 1000 | 2000 | 3000 | 4000 | 6000 | 8000 |
| Nira | Acc | 87.74 | **87.89** | 87.67 | 87.26 | 86.69 | 87.75 | 87.61 |
| | Atk Rate | 1.6 | 3.22 | 2.23 | **1.18** | 3.17 | 2.23 | 3.26 |
| | Main Acc | 87.84 | **88.01** | 87.88 | 87.27 | 86.89 | 87.87 | 87.81 |

Table 18: Performance of different defense algorithms without the scale-up operation.

| Defense | Atk Num = 0 | | | Atk Num = 10 | | | Atk Num = 20 | | | Atk Num = 30 | | |
|---|---|---|---|---|---|---|---|---|---|---|---|---|
| | Acc | Atk Rate | Main Acc | Acc | Atk Rate | Main Acc | Acc | Atk Rate | Main Acc | Acc | Atk Rate | Main Acc |
| FedAvg | 89.35 | **0.67** | 89.5 | 81.99 | 90.94 | **89.37** | 82.02 | 91.68 | **89.46** | 81.7 | 95.72 | **89.49** |
| RFA | 87.51 | 1.16 | 87.56 | 80.19 | 89.18 | 87.24 | 80.1 | 90.78 | 87.25 | 79.48 | 93.59 | 86.88 |
| MultiKrum | 88.33 | 1.22 | 88.4 | 81.05 | 90.28 | 88.27 | 81.05 | 91.42 | 88.38 | 81.05 | 93.09 | 88.53 |
| Coomed | 87.91 | 0.96 | 87.91 | 80.57 | 88.72 | 87.64 | 80.4 | 90.98 | 87.65 | 79.85 | 92.25 | 87.17 |
| Normclip | 79.71 | 1 | 79.65 | 72.05 | 82.47 | 77.81 | 71.61 | 87.1 | 77.7 | 71.57 | 89.67 | 77.82 |
| Nira(ours) | **89.74** | 0.94 | **89.78** | **83.65** | 38.8 | 87.13 | **84.21** | 39.2 | 87.42 | **84.06** | 43.8 | 87.78 |

Table 19: Results of only one attacker and no attackers under different poison ratios for both FedAvg and Nira.

| Metric | Defense | 0 attacker | Poison Ratios (1 attacker) | | | | | | |
|---|---|---|---|---|---|---|---|---|---|
| | | | 1% | 5% | 10% | 20% | 30% | 40% | 50% |
| Acc | FedAvg | 89.35 | 64.16 | 75.69 | 72.14 | 73.88 | 70.04 | 70.25 | 66.09 |
| | Nira(ours) | **89.74** | **87.2** | **89.03** | **87.96** | **89.18** | **88.66** | **88.69** | **88.01** |
| Atk Rate | FedAvg | **0.67** | 1.53 | 89.07 | 92.47 | 97.12 | 93.96 | 93.59 | 93.99 |
| | Nira(ours) | 0.94 | **0.67** | **1.11** | **1.2** | **0.41** | **0.54** | **1.07** | **0.92** |
| Main Acc | FedAvg | 89.5 | 64.16 | 82.43 | 78.69 | 81.02 | 76.54 | 76.74 | 72.2 |
| | Nira(ours) | **89.78** | **87.24** | **89.19** | **87.99** | **89.17** | **88.77** | **88.76** | **88.01** |

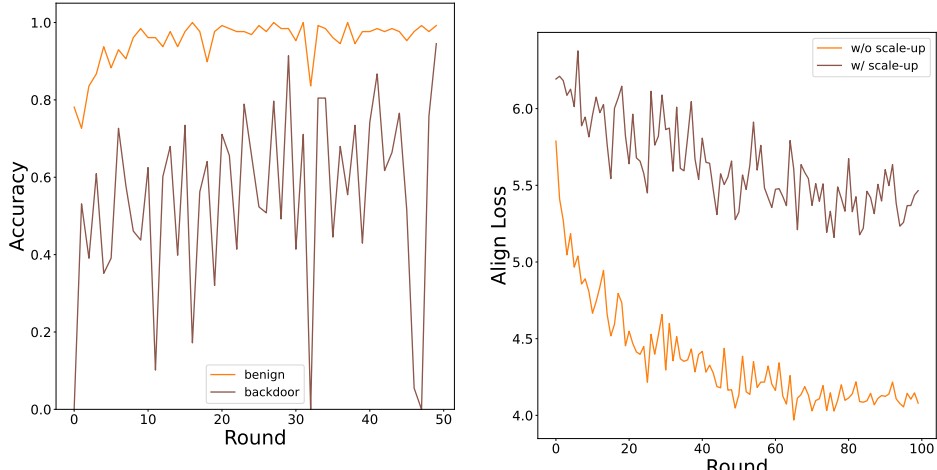

Figure 8: The accuracy of a benign client and malicious client. The benign data of both clients is fixed to be the same while introducing additional poisoned samples to the malicious client.

Figure 9: The alignment loss between poisoned data and surrogate data with (without) the scale-up operation.

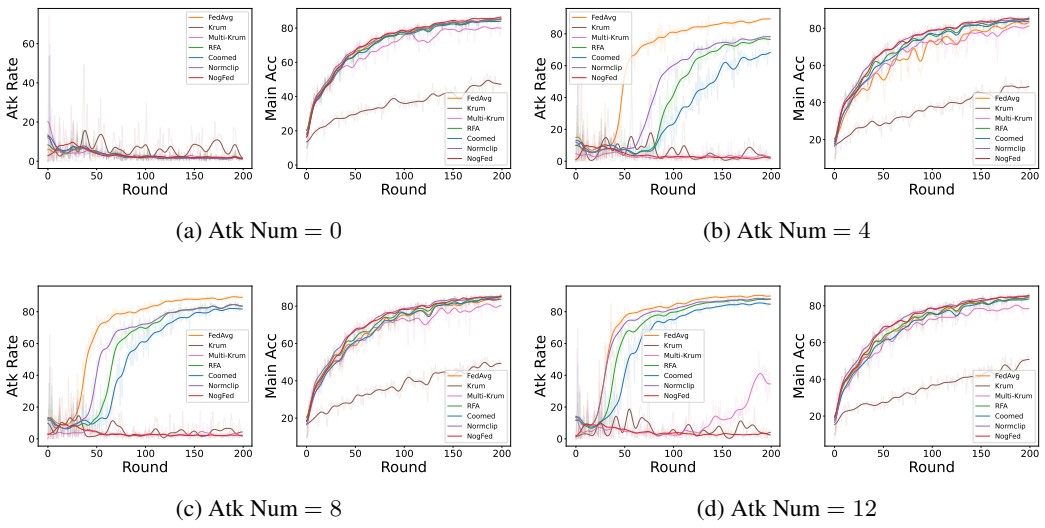

Figure 10: Curves of Atk Rate and Main Acc on CIFAR-10.

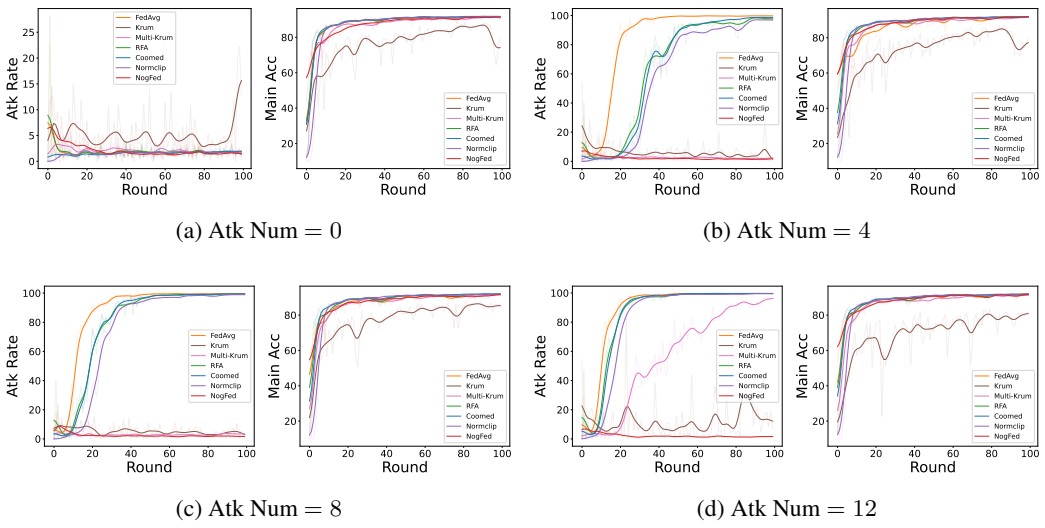

Figure 11: Curves of Atk Rate and Main Acc on FMNIST.

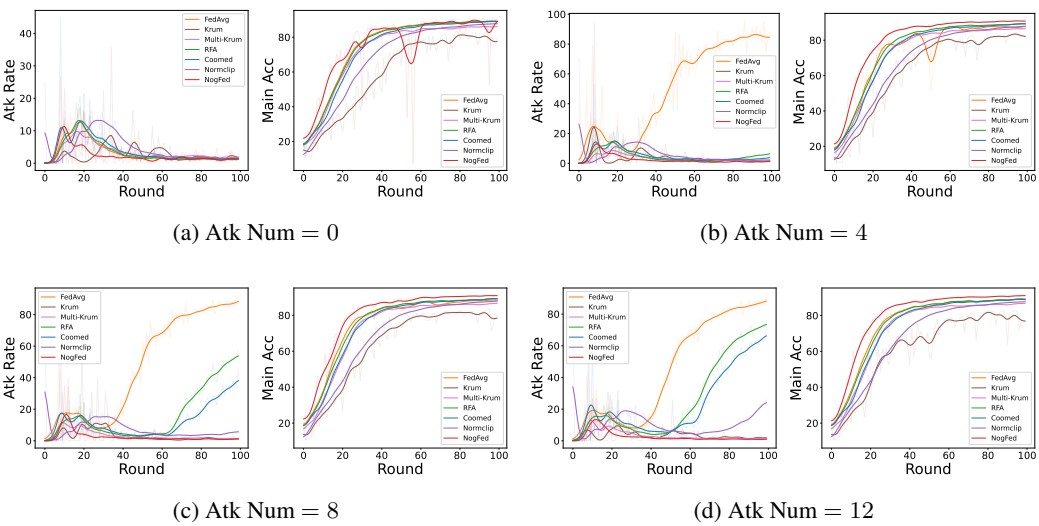

Figure 12: Curves of Atk Rate and Main Acc on SVHN.

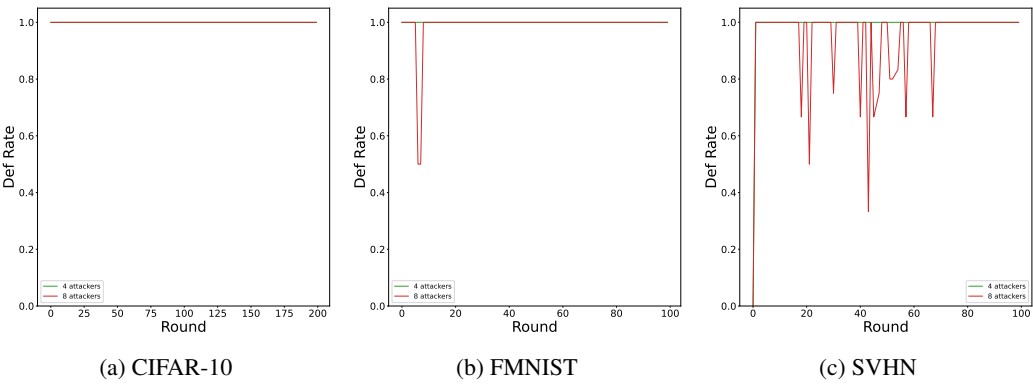

Figure 13: Curves of Defense Rate on CIFAR-10, FMNIST and SVHN.

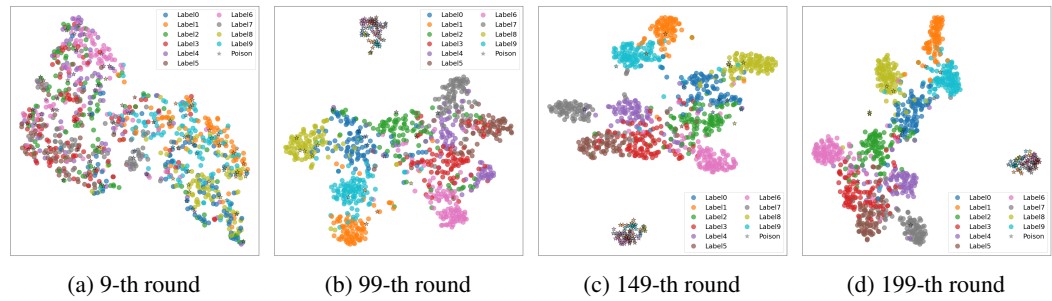

(a) 9-th round      (b) 99-th round      (c) 149-th round      (d) 199-th round

Figure 14: Features of **CIFAR-10** test data of server model with FedAvg

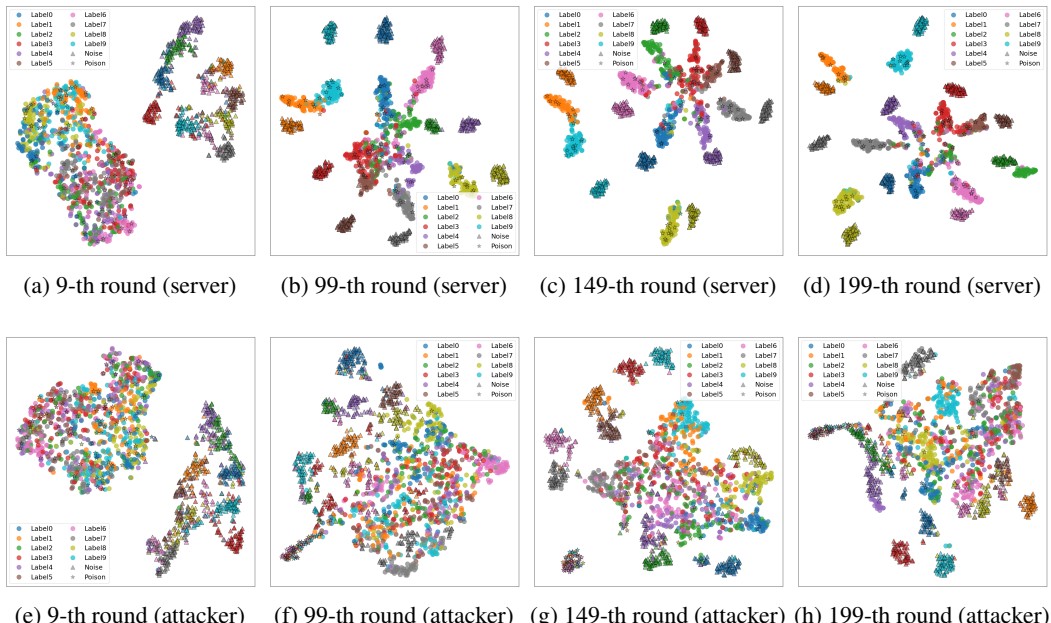

(a) 9-th round (server)    (b) 99-th round (server)    (c) 149-th round (server)    (d) 199-th round (server)

(e) 9-th round (attacker)    (f) 99-th round (attacker)    (g) 149-th round (attacker)    (h) 199-th round (attacker)

Figure 15: Features of **CIFAR-10** test data with Nira, with $\alpha = 1$

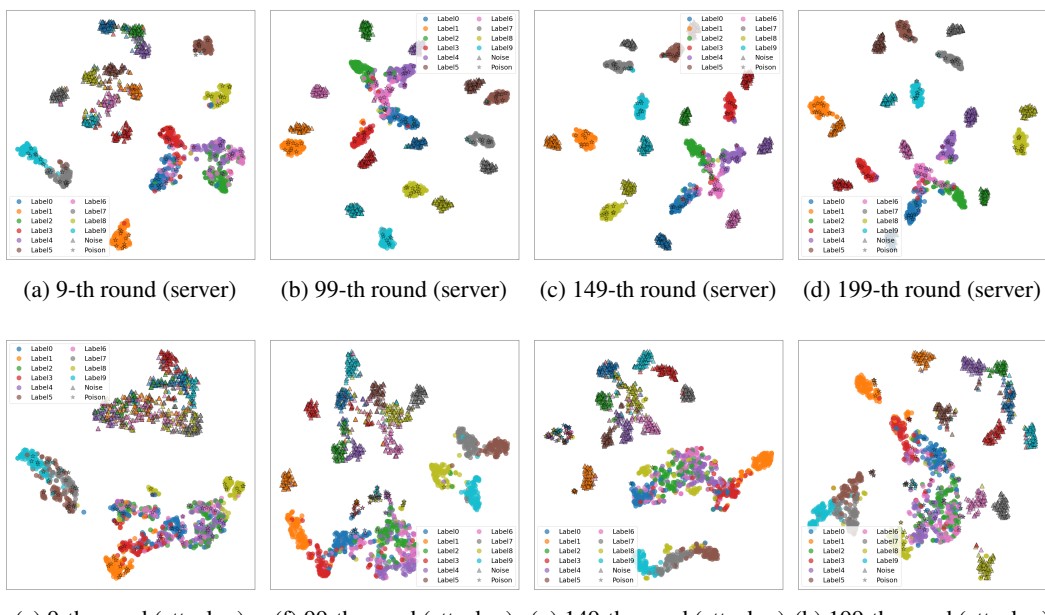

(a) 9-th round (server)  (b) 99-th round (server)  (c) 149-th round (server)  (d) 199-th round (server)

(e) 9-th round (attacker)  (f) 99-th round (attacker)  (g) 149-th round (attacker)  (h) 199-th round (attacker)

Figure 16: Features of **FMNIST** test data with Nira, with $\alpha = 1$

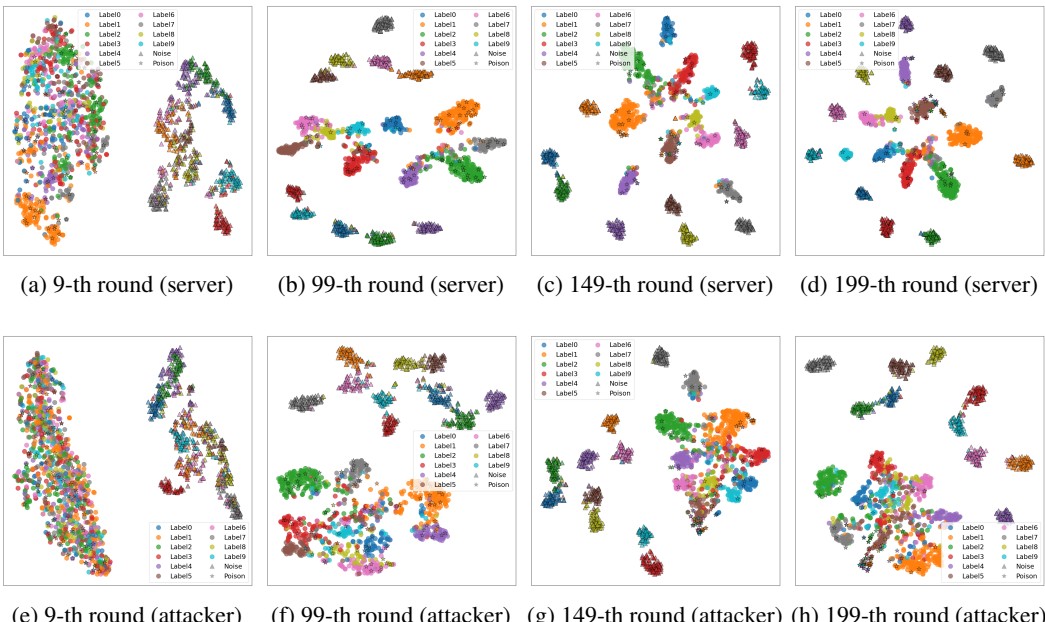

(a) 9-th round (server)  (b) 99-th round (server)  (c) 149-th round (server)  (d) 199-th round (server)

(e) 9-th round (attacker)  (f) 99-th round (attacker)  (g) 149-th round (attacker)  (h) 199-th round (attacker)

Figure 17: Features of **SVHN** test data with Nira, with $\alpha = 1$

