# OpenReview forum: "Mitigating Backdoor Attacks in Federated Learning through Noise-Guided Aggregation"
_ICLR.cc/2024/Conference — Submitted to ICLR 2024_

### Official Review · Reviewer_yMFU · 2023-10-18

**Soundness:** 3 good
**Presentation:** 2 fair
**Contribution:** 3 good
**Rating:** 6
**Confidence:** 4

**Summary:**

This paper proposes Nira, a backdoor defense for federated learning based on the idea of noise-guided robust aggregation.

**Strengths:**

* The proposed noise-guided defense is novel among the defense literature.

* Unlike many existing defenses, the method does not require a surrogate dataset (but only noise) in filtering out the poisoned model from the aggregation phase.

* The benign accuracy is increased due to alignment with noise, while for most existing defenses, the accuracy is suffered.

**Weaknesses:**

* The reason for the effectiveness of noise alignment and filtering lacks clarity. It remains uncertain why attackers struggle to align their data with noise. It is advisable for the authors to conduct a toy experiment to illustrate this point. For instance, in a scenario involving two clients, we could fix the benign data of both clients to be the same while introducing additional poisoned samples to the first client. Performing one epoch of noise alignment with these two clients and comparing their accuracy on noise samples would shed light on this issue.


* The provided rationale for Nira's ability to defend against adaptive attacks is not clear too. The authors shows that Nira is robust to hte adaptive attack where attackers split their poisoned dataset into poisoned and benign segments, aligning only the surrogate samples with the data within the benign segments. The reasons given are that 1) "scale-up operation on the model weights has its impact" and 2) poisoned samples in the cross-entropy loss lead to poor alignment. It is suggested that the authors i) disable the scale-up operation, and ii) in the absence of the scale-up operation, substantiate the second reason with experimental results. This is important because it might not be reasonable to assume that attackers cannot produce an over-parameterized model that can simultaneously minimize the alignment loss and the backdoor cross-entropy loss, as these objectives are not inherently contradictory.

**Questions:**

The statement that "calculating these metrics involves malicious models, leading to biased metrics and defense failure" suggests that involving malicious models in metric calculation introduces bias and can result in defense failure. However, this statement needs further clarification, and it might be beneficial for the authors to elaborate on the specific reasons why using malicious models for metric calculation could lead to biased results and ultimately undermine the defense. Additionally, it's worth addressing why Nira's use of malicious models in a similar context does not result in failure.

In Table 2, the impact of poisoning ratio on benign accuracy appears significant, which is not a usual case for backdoor attack. To better understand this phenomenon, it would be helpful to see the results for scenarios with only one attacker (attacker number = 1) and no attackers (attacker number = 0) under different poisoning ratios for both FedAvg and Nira. This would provide a clearer picture of the algorithm's performance under varying conditions.

It is suggested to provide experimental results when full client participation is employed, without client selection. The randomness associated with client selection could indeed influence training results, particularly affecting the ASR. Demonstrating how the approach performs under full participation conditions would offer insights into its stability.

Clarification on the scale factor in replacement attacks is necessary. It would also be valuable to determine whether Nira remains effective when the scaling-up operation is disabled. Note that the scaling-up operation may not be essential for conducting backdoor attacks.

I would consider adjusting my score based on the authors rebuttal, and will actively participate in it.

---

> ### Author Response · Authors · 2023-11-18
> **Response to Reviewer yMFU (1/3)**
>
> We would like to thank the reviewer for taking the time to review. We appreciate that you find that our method is novel, only requires noise, and improves benign accuracy by aligning with noise. In light of your insightful comments, we offer comprehensive responses below. We also add them into our main text or appendix in the revision. We hope that these responses and the revision have effectively resolved your concerns and enhanced the overall excellence of our work.
>
> > **Q1:** The reason for the effectiveness of noise alignment and filtering lacks clarity. It remains uncertain why attackers struggle to align their data with noise. It is advisable for the authors to conduct a toy experiment to illustrate this point.
>
> ***Ans for Q1):*** Thanks for your valuable comments.
>
> - We exploited t-SNE to visualize the feature distribution of the server and clients in Figures 4, 13, 14, and 15. It can be observed that the attacker’s feature distribution is more scattered and loose compared to the benign client, making it perform worse on the surrogate noise data.
> - In response to your constructive comments, we have conducted toy experiments involving two clients. The benign data of both clients is the same while introducing additional poisoned samples to the first client. The accuracy of these two clients on noise samples is shown in Figure 8 Appendix E.11. We can see that the client introducing poisoned samples performs worse on noise samples, making it easier for the server to identify attackers.
>
> We have added the above results and discussion to our revision.
>
>
> > **Q2:** The provided rationale for Nira's ability to defend against adaptive attacks is not clear too. It is suggested that the authors i) disable the scale-up operation, and ii) in the absence of the scale-up operation, substantiate the second reason with experimental results.
>
> ***Ans for Q2):*** Thanks for your constructive comments.
>
> - We apologize for the lack of detailed analysis and experimental support for Nira's defense against adaptive attacks. According to your valuable comments, we analyze the correlation between poisoned data, benign data, and surrogate data below. When attackers train local models, they align the conditional feature distributions between benign data and surrogate data to circumvent the server's detection. In this context, aligning the conditional feature distributions of poisoned data and surrogate data would also align the distributions of poisoned data and benign data, thereby reducing the damage of poisoned samples. If poisoned data and surrogate data are not aligned, the model's accuracy on surrogate samples will be low, making it easy for the server to detect. Therefore, the proposed method can effectively defend against malicious attacks whether poisoned data is aligned with surrogate data or not.
> - To further validate the impact of scale-up on the performance of Nira, we conduct experiments without the scale-up operation, and the alignment loss between poisoned data and surrogate data is shown in Figure 9 Appendix E.11. We can see that the scale-up operation increases the alignment loss between poisoned data and surrogate data.
>
> We have added the above results and discussion to our revision.
>
> > **Q3:** It might be beneficial for the authors to elaborate on the specific reasons why using malicious models for metric calculation could lead to biased results and ultimately undermine the defense. Additionally, it's worth addressing why Nira's use of malicious models in a similar context does not result in failure.
>
> ***Ans for Q3):*** We appreciate your suggestion. We have added detailed explanations in the revised paper.
>
> - In previous methods, malicious models are also involved in the calculation of these metrics, thus may yield tainted metrics and fail to achieve effective defense. For example, Krum [1] calculates the sum of the squared distances between each client model and the other client models as its score, and aggregates several models with the lowest scores. However, when the majority are attackers, the score of the malicious model may be relatively lower, leading the server to aggregate malicious models and resulting in defense failure.
> - The proposed method aims to defend against backdoor attacks by designing a metric that will not be tainted by malicious models. To this end, we propose a metric that performs individual evaluation for each local model using surrogate data. This individual evaluation approach renders the metric impervious to variations in the attacker's ratio.
> - The proposed metric (accuracy on surrogate data) can successfully filter out most malicious models, which is consistent with our experimental results reported in Table 5 Appendix E.4.

---

> ### Author Response · Authors · 2023-11-18
> **Response to Reviewer yMFU (2/3)**
>
> > **Q4:** In Table 2, the impact of the poisoning ratio on benign accuracy appears significant, which is not a usual case for backdoor attacks. To better understand this phenomenon, it would be helpful to see the results for scenarios with only one attacker (attacker number = 1) and no attackers (attacker number = 0) under different poisoning ratios for both FedAvg and Nira.
>
> ***Ans for Q4):*** Thanks for pointing out this potentially confusing problem.
>
> - In Table 2, "Acc" refers to the prediction accuracy computed on the clean and poisoned test samples; "Main Acc" refers to the prediction accuracy computed on clean test samples. We can see that the poison ratio has a limited impact on the "Main Acc".
> - In response to your constructive comments, we have conducted experiments with only one attacker (attacker number = 1) and no attackers (attacker number = 0) under different poison ratios for both FedAvg and Nira. The experimental results are shown in Table 3-1 as below. We can see that the proposed method achieves good performance under varying poison ratios.
>
> **Table 3-1**
> |Metric|Defense|0 attacker||||Poison Ratios (1 attacker)||||
> |:-:|:-:|:-:|:-:|:-:|:-:|:-:|:-:|:-:|:-:|
> ||||1%|5%|10%|20%|30%|40%|50%|
> |Acc|FedAvg|89.35|64.16|75.69|72.14|73.88|70.04|70.25|66.09|
> ||Nira(ours)|**89.74**|**87.2**|**89.03**|**87.96**|**89.18**|**88.66**|**88.69**|**88.01**|
> |----|----|----|----|----|----|----|----|----|----|
> |Atk Rate|FedAvg|**0.67**|1.53|89.07|92.47|97.12|93.96|93.59|93.99|
> ||Nira(ours)|0.94|**0.67**|**1.11**|**1.2**|**0.41**|**0.54**|**1.07**|**0.92**|
> |----|----|----|----|----|----|----|----|----|----|
> |Main Acc|FedAvg|89.5|64.16|82.43|78.69|81.02|76.54|76.74|72.2|
> ||Nira(ours)|**89.78**|**87.24**|**89.19**|**87.99**|**89.17**|**88.77**|**88.76**|**88.01**|
>
> Following your valuable comments, we have added the above results and discussion in our revision.
>
>
> > **Q5:** It is suggested to provide experimental results when full client participation is employed, without client selection.
>
> ***Ans for Q5):*** Thanks for your suggestion, which motivates the following experiments. We have added the following experiments without client selection to our revision.
>
> - We have conducted experiments using a larger ratio, $30/50=60\%$. The results are reported in Table 3-2 as below. We can see that the proposed method achieves good performance even when attackers constitute the majority.
>
> **Table 3-2**
> |Defense||Atk Num = 0|||Atk Num = 10|||Atk Num = 20|||Atk Num = 30||
> |:-:|:-:|:-:|:-:|:-:|:-:|:-:|:-:|:-:|:-:|:-:|:-:|:-:|
> ||Acc|Atk Rate|Main Acc|Acc|Atk Rate|Main Acc|Acc|Atk Rate|Main Acc|Acc|Atk Rate|Main Acc|
> |FedAvg|89.35|**0.67**|89.5|79.46|97.23|87.17|81.61|94.82|**89.3**|81.99|96.92|**89.94**|
> |RFA|87.51|1.16|87.56|80.83|81.25|87.24|80.4|86.75|87.23|79.67|91.88|86.91|
> |MultiKrum|88.33|1.22|88.4|88.32|1.35|88.47|80.35|88.31|87.3|81.03|97.01|88.88|
> |Coomed|87.91|0.96|87.91|81.41|89.1|88.57|80.94|91.51|88.27|80.64|95.46|88.3|
> |Normclip|79.71|1|79.65|73.8|83.57|79.77|73.16|87.8|79.43|73.26|89.8|79.68|
> |Nira(ours)|**89.74**|0.94|**89.78**|**88.85**|**1.24**|**88.92**|**87.67**|**2.23**|87.88|**85.13**|**3.46**|85.38|
>
> - We have conducted experiments using larger ratios, $40\%, 60\%$, and $80\%$. The results are reported in Table 3-3 as below. We can see that attackers fail to enhance attack performance by increasing the poison ratio.
>
> **Table 3-3**
> |Defense||Poison Ratio = 20%|||Poison Ratio = 40%|||Poison Ratio = 60%|||Poison Ratio = 80%||
> |:-:|:-:|:-:|:-:|:-:|:-:|:-:|:-:|:-:|:-:|:-:|:-:|:-:|
> ||Acc|Atk Rate|Main Acc|Acc|Atk Rate|Main Acc|Acc|Atk Rate|Main Acc|Acc|Atk Rate|Main Acc|
> |FedAvg|81.1|99.25|**89.16**|80.48|99.75|**88.53**|79.05|99.69|**86.95**|76.92|99.78|84.62|
> |RFA|79.44|89.51|86.71|78.12|88.55|85.07|75.95|92.21|83.03|74.95|85.54|81.54|
> |MultiKrum|80.36|99.47|88.37|79.9|99.75|87.9|77.88|99.69|85.67|74.92|99.78|82.42|
> |Coomed|80.28|98.04|88.15|79.67|99.32|87.6|78.29|99.78|86.13|75.01|99.75|82.51|
> |Normclip|71.92|92.34|78.45|69.27|93.13|75.62|66.38|93.55|72.48|61.66|93.79|67.31|
> |Nira(ours)|**84.78**|**4.34**|85.09|**84.9**|**4.58**|85.16|**85.02**|**3.92**|85.23|**84.8**|**4.4**|**85.09**|
>
> In sum, experiments with larger ratios of malicious clients and poisons further underscore the robust effectiveness of the proposed method in diverse settings. Thanks again for your constructive comments.

---

> ### Author Response · Authors · 2023-11-18
> **Response to Reviewer yMFU (3/3)**
>
> > **Q6:** Clarification on the scale factor in replacement attacks is necessary. It would also be valuable to determine whether Nira remains effective when the scaling-up operation is disabled.
>
> ***Ans for Q6):*** Thanks for your valuable comments. We have added the following clarification and experiments to our revision.
>
> - The scale factor we used is $\frac{m}{n_a}$, where $m$ is the number of clients participating in aggregation each round and $n_a$ is the number of attackers, which is consistent with previous outstanding works [2,3].
> - In response to your constructive comments, we have conducted experiments without the scale-up operation. The results are reported in Table 3-4 as below. We can see that the proposed method still outperforms baselines in the absence of the scale-up operation.
>
> **Table 3-4**
> |Defense||Atk Num = 0|||Atk Num = 10|||Atk Num = 20|||Atk Num = 30||
> |:-:|:-:|:-:|:-:|:-:|:-:|:-:|:-:|:-:|:-:|:-:|:-:|:-:|
> ||Acc|Atk Rate|Main Acc|Acc|Atk Rate|Main Acc|Acc|Atk Rate|Main Acc|Acc|Atk Rate|Main Acc|
> |FedAvg|89.35|**0.67**|89.5|81.99|90.94|**89.37**|82.02|91.68|**89.46**|81.7|95.72|**89.49**|
> |RFA|87.51|1.16|87.56|80.19|89.18|87.24|80.1|90.78|87.25|79.48|93.59|86.88|
> |MultiKrum|88.33|1.22|88.4|81.05|90.28|88.27|81.05|91.42|88.38|81.05|93.09|88.53|
> |Coomed|87.91|0.96|87.91|80.57|88.72|87.64|80.4|90.98|87.65|79.85|92.25|87.17|
> |Normclip|79.71|1|79.65|72.05|82.47|77.81|71.61|87.1|77.7|71.57|89.67|77.82|
> |Nira(ours)|**89.74**|0.94|**89.78**|**83.65**|**38.8**|87.13|**84.21**|**39.2**|87.42|**84.06**|**43.8**|87.78|
>
>
> > ***Reference***
> >
> > [1] Blanchard, Peva, et al. "Machine learning with adversaries: Byzantine tolerant gradient descent." Advances in neural information processing systems 30 (2017).
> >
> > [2] Bagdasaryan, Eugene, et al. "How to backdoor federated learning." International conference on artificial intelligence and statistics. PMLR, 2020.
> >
> > [3] Wang, Hongyi, et al. "Attack of the tails: Yes, you really can backdoor federated learning." Advances in Neural Information Processing Systems 33 (2020): 16070-16084.

---

> ### Comment · Reviewer_yMFU · 2023-11-18
> **Thanks for the rebuttal, but my concern remains.**
>
> I am still concerning about your answer on Q2. Under the adaptive attack scenario, the attacker only aligns the benign data with surrogate data. But why it is claimed that "aligning the conditional feature distributions of poisoned data and surrogate data would also align the distributions of poisoned data and benign data, thereby reducing the damage of poisoned sample". Attacker is not going to "align the conditional feature distributions of poisoned data and surrogate data".  The logic does not make sense here.

---

> > ### Author Response · Authors · 2023-11-18
> > **Response to Reviewer yMFU**
> >
> > We sincerely thank you for your prompt feedback! We would like to provide more detailed explanations for your insightful question.
> >
> > In the adaptive attack scenario, attackers will merely align benign data with surrogate data. The intuition is that the adaptive attack may degenerate the performance of our method. However, our method still works well. The insight into this phenomenon is attractive. Before clarifying the picture, we should ensure our perspective aligns with yours on the problem.
> >
> > - i) Aligning the benign samples with surrogate samples will contribute to the prediction accuracy of benign and surrogate samples.
> > - ii) Updating models with poisoning samples will degenerate the prediction accuracy of benign samples.
> >
> > Built upon the above two points, we can derive that updating models with poisoning samples will degenerate the prediction accuracy of surrogate samples. This is because surrogate samples are aligned with the benign samples in the feature space. Thus, even if poisoning samples are not leveraged to be aligned with surrogate samples by the attackers, the performance on surrogate data will still degenerate. As a result, our method can detect malicious models using the prediction accuracy of surrogate samples.
> >
> > We will add the above discussion to our revision. We would like to express that we enjoy the in-depth discussion with you, so if you have any further outstanding comments or questions, please do not hesitate to post them.

---

> > > ### Comment · Reviewer_yMFU · 2023-11-18
> > > **I could not  agree with your second claim**
> > >
> > > It is claimed that  "updating models with poisoning samples will degenerate the prediction accuracy of benign samples". This does not necessarily hold for backdoor attack if the poison ratio is set to be small (See Table 2 in [A]). For a successsful backdoor attack, the prediction accuracy of benign samples should not suffer much.
> > >
> > > Also, there are ways that the attacker can boost the prediction accuracy of benign samples (compared to other benign clients), e.g.,  increasing the attacker's local training epochs.
> > >
> > >
> > > [A] Tran B, Li J, Madry A. Spectral signatures in backdoor attacks[J]. Advances in neural information processing systems, 2018, 31.

---

> > > > ### Author Response · Authors · 2023-11-19
> > > > **Response to Reviewer yMFU**
> > > >
> > > > We sincerely thank you for your prompt feedback! We would like to apologize for the previous inaccurate response. We are currently unable to determine the reasons why the proposed method remains effective under the adaptive attack. We will add this information in the revised version once the underlying reasons are identified.

---

### Official Review · Reviewer_nxZi · 2023-10-26

**Soundness:** 3 good
**Presentation:** 3 good
**Contribution:** 2 fair
**Rating:** 6
**Confidence:** 3

**Summary:**

The paper proposes a new method to defend against backdoor attacks in federated learning. A surrogate dataset is generated based on Gaussian noise to help detect malicious clients. The surrogate dataset will be shared to clients and the clients will train models based on both local data and the surrogate data to get models that perform well on both local data and the surrogate data. Malicious clients are detected based on measuring accuracy and feature distances of local models on the surrogate data. Experiments on vision data show that the proposed method performs well under certain scenarios.

**Strengths:**

Strengths:

- The proposed method does not need the assumption that the majority of the clients are benign. It also does not need access to an auxiliary dataset which some previous methods require.

- Experiments show that the proposed method performs well on three vision datasets.

- The paper is clearly written with several graphs illustrating the idea and framework.

- Backdoor attack is an important security issue in the field of federated learning.

**Weaknesses:**

Weaknesses:

- The paper claims that it is not majority-based defense. Based on the description, it seems right. However, why not evaluate the claim with experiments? There are 50 clients in the experiment, and the number of attackers ranges from 0 to 12, which corresponds to 0% to 24% of malicious clients, not exceeding 50%.

- The thresholds of filtering clients seem to be important hyper-parameters. In the experimental part, it is explained that the accuracy threshold increases as the training rounds progress and the distance threshold decreases. The specific thresholds in different rounds are also given. However, details of threshold selection are not included in the main paper. In the appendix, it says that "train a few rounds on a small number of identified benign clients". If the thresholds are selected this way, knowing some identified benign clients is an important assumption of the method that should be mentioned in the main paper.

- This method seems to be limited to vision data. Federated learning can also be applied to other types of data, such as NLP, graph. How could the method generalize to other types of data?

**Questions:**

See previous part.

---

> ### Author Response · Authors · 2023-11-18
> **Response to Reviewer nxZi**
>
> We sincerely thank the reviewer for taking the time to review. We appreciate that you find that the backdoor attack is an important security issue in FL, the paper is clearly written, and the proposed method does not need the assumption that the majority of the clients are benign and does not need access to an auxiliary dataset which some previous methods require. According to your insightful and valuable comments, we provide detailed feedback below and add them into our main text or appendix in the revision. We hope these modifications and our revision can address your concerns and improve our work.
>
>
> > **Q1:** There are 50 clients in the experiment, and the number of attackers ranges from 0 to 12, which corresponds to 0% to 24% of malicious clients, not exceeding 50%.
>
> ***Ans for Q1):*** We appreciate your valuable comments, which motivate the following experiments. We have added the following experiments to our revision.
>
> In response to your constructive comments, we have conducted experiments using a larger ratio, $30/50=60\%$. The results are reported in Table 2-1. We can see that the proposed method achieves good performance even when attackers constitute the majority, which further underscores the robust effectiveness of the proposed method.
>
> **Table 2-1**
> |Defense||Atk Num = 0|||Atk Num = 10|||Atk Num = 20|||Atk Num = 30||
> |:-:|:-:|:-:|:-:|:-:|:-:|:-:|:-:|:-:|:-:|:-:|:-:|:-:|
> ||Acc|Atk Rate|Main Acc|Acc|Atk Rate|Main Acc|Acc|Atk Rate|Main Acc|Acc|Atk Rate|Main Acc|
> |FedAvg|89.35|**0.67**|89.5|79.46|97.23|87.17|81.61|94.82|**89.3**|81.99|96.92|**89.94**|
> |RFA|87.51|1.16|87.56|80.83|81.25|87.24|80.4|86.75|87.23|79.67|91.88|86.91|
> |MultiKrum|88.33|1.22|88.4|88.32|1.35|88.47|80.35|88.31|87.3|81.03|97.01|88.88|
> |Coomed|87.91|0.96|87.91|81.41|89.1|88.57|80.94|91.51|88.27|80.64|95.46|88.3|
> |Normclip|79.71|1|79.65|73.8|83.57|79.77|73.16|87.8|79.43|73.26|89.8|79.68|
> |Nira(ours)|**89.74**|0.94|**89.78**|**88.85**|**1.24**|**88.92**|**87.67**|**2.23**|87.88|**85.13**|**3.46**|85.38|
>
>
> > **Q2:** In the appendix, it says that "train a few rounds on a small number of identified benign clients". If the thresholds are selected this way, knowing some identified benign clients is an important assumption of the method that should be mentioned in the main paper.
>
> ***Ans for Q2):*** Thanks for pointing out the lack of information about the assumption. To efficiently select thresholds, we need to assume that the server can identify a small number of benign clients and simulate the training process. Following your valuable comments, we have highlighted this assumption in our revision.
>
> > **Q3:** This method seems to be limited to vision data. Federated learning can also be applied to other types of data, such as NLP, and graphs. How could the method generalize to other types of data?
>
> ***Ans for Q3):*** Thanks for your constructive comments.
>
> - Extending our proposed method to different data types, including NLP and graphs, is an interesting question.
> - The promising direction may align with the work [1]. However, due to time limitations, we leave it as future work.
>
>
> > ***Reference***
> >
> > [1] Baradad Jurjo, Manel, et al. "Learning to see by looking at noise." Advances in Neural Information Processing Systems 34 (2021): 2556-2569.

---

### Official Review · Reviewer_ig5G · 2023-10-31

**Soundness:** 2 fair
**Presentation:** 3 good
**Contribution:** 2 fair
**Rating:** 3
**Confidence:** 5

**Summary:**

This paper proposed a novel approach utilizing generated (surrogate) dataset for mitigating the backdoor attacks in federated learning.  Domain adaption is performed to align the distribution between the surrogate and original dataset. Experimental results demonstrate the effectiveness of the approach under various settings.

**Strengths:**

1. The proposed approach is technically novel and sound.
2. The paper is well-written and easy to follow in general.
3. Experimental evaluations are comprehensive, covering a broad aspect of factors and baseline methods.

**Weaknesses:**

1. The proposed approach is based on strong assumptions that the malicious attackers follow the proposed protocol, i.e. training both surrogate and local dataset with the proposed objectives exactly or adaptively (in Adaptive Attack Scenario). However, the introduction of surrogate data may easily prompt the malicious attackers to inject backdoor triggers and objectives to the surrogate dataset as well. Have the author considered these scenarios?

2. The paper claims that "We point out that metrics used for backdoor defense can be tainted by malicious models, leading to the failure of existing approaches." as its contribution, however, it is not convincing that the proposed method adequately addressed this problem since 1) the metric that server used for filtering out malicious attackers appear to use a distance outliner (eq.6) and thresholds, which may be also tainted if the majority are attackers. 2) experimental results are limited to small ratio of clients (12/50) and poisons (20%).

3. Missing Details. The design of the filtering thresholds is crucial to the proposed method but is not explained well. Why they vary with time in experiments? Are there any insights on how to adjust them systematically? In 3.3, what does "features of the model" mean and how to determine the value of b? In Table 1, the definition of "Acc" , "Atk Rate" and "Main Acc" are not explicitly explained and therefore confusing.

4. The experiments are conducted on relatively limited settings, i.e., only one triggered-based backdoor attack is considered, and small dataset.

**Questions:**

1. What is the ratio of the number of surrogate data and real data for each clients? How will this ratio affect the performance?

---

> ### Author Response · Authors · 2023-11-18
> **Response to Reviewer ig5G (1/3)**
>
> We sincerely thank the reviewer for taking the time to review. We appreciate that you find that our method is technically novel and sound, our paper is well-written and easy to follow, and the experimental evaluations are comprehensive. According to your valuable comments, we provide detailed feedback below and also add them into our main text or appendix in the revision. We hope that these responses have addressed your concerns and improved our work.
>
> > **Q1:** The proposed approach is based on strong assumptions that the malicious attackers follow the proposed protocol, i.e. training both surrogate and local datasets with the proposed objectives exactly or adaptively.
>
> ***Ans for Q1):*** Thanks for your insightful comments. We agree with the point that malicious attackers may reject to follow the proposed protocol. In this context, the model sent from malicious attackers will produce a poor performance on the constructed surrogate data. Accordingly, our method will detect these models as malicious models. Therefore, malicious attackers who reject to follow the proposed protocol fail to mount a backdoor attack.
>
> We have highlighted this in-depth discussion in our revision. Thanks again for your constructive comments.
>
>
> > **Q2:** The introduction of surrogate data may easily prompt malicious attackers to inject backdoor triggers and objectives into the surrogate dataset as well. Have the authors considered these scenarios?
>
> ***Ans for Q2):*** We apologize for the missing discussion. Following your valuable question, we have added the following experiments and discussion in our revision.
>
> Malicious attacks may inject backdoor triggers into the surrogate data. However, if an attacker attempts to mount a backdoor attack on the surrogate data, the model sent from the attacker will achieve poor performance on the surrogate data shared among clients and server. As a result, the server will filter out these malicious models by evaluating their performance on the surrogate data. Therefore, injecting backdoor triggers on the surrogate data will also be detected by the proposed method. To verify the point, we inject backdoor triggers into the surrogate dataset on a malicious client while other settings are the same as our original setting. The results are reported in Table 1-1 as below. The results show that these malicious clients will not bypass the proposed method.
>
> **Table 1-1**
> |Defense|Metric||||Surrogate Data Poison Ratio||||
> |:-:|:-:|:-:|:-:|:-:|:-:|:-:|:-:|:-:|
> |||0%|5%|10%|20%|40%|60%|80%|
> ||Acc|87.67|87.59|**88**|87.78|87.41|87.54|87.02|
> |Nira|Atk Rate|**2.23**|2.23|2.41|2.65|3.09|2.47|3.15|
> ||Main Acc|87.88|87.83|**88.22**|88.05|87.79|87.82|87.22|
>
> > **Q3:** The metric that the server used for filtering out malicious attackers appears to use a distance outliner (eq.6) and thresholds, which may be also tainted if the majority are attackers.
>
> ***Ans for Q3):*** Thanks for pointing out the potentially confusing description. We have added detailed explanations in the revised paper.
>
> - The proposed method aims to defend against backdoor attacks by designing a metric that will not be tainted by malicious models. To this end, we propose a metric that performs individual evaluation for each local model using surrogate data. This individual evaluation approach renders the metric impervious to variations in the attacker's ratio.
> - The proposed metric (accuracy on surrogate data) can successfully filter out most malicious models, which is consistent with our experimental results reported in Table 5 Appendix E.4.
> - However, it is challenging for a single metric to detect all malicious models, even if the metric is not tainted by malicious models. Thus, we introduce a supplementary metric to further promote the performance. The supplementary metric is inspired by existing methods, employing the distance among local models. Thus, the supplementary metric has the risk of being tainted.
> - By integrating these two metrics, the proposed method achieves a more potent defense performance, particularly in scenarios with a significant proportion of attackers.

---

> > ### Comment · Reviewer_ig5G · 2023-11-22
> >
> > I have read the authors' responses and I appreciated the new results and discussions, which have addressed some of my concerns. However, my biggest concern is still valid, which is,  using a surrogate dataset and heuristic filtering metrics can be easily circumvent by adaptive attackers. I am not convinced by the results on further backdooring surrogate dataset, partly because it is not clear if this attack is adequately implemented. Knowing the filtering metrics, the backdoor attacker can inject this into its objectives by insuring the model accuracy of both surrogate dataset and backdoored task, just as a common backdoor task. Therefore it is still unclear to me how the model accuracy of surrogated dataset are still compromised by large margins in these circumstances.  Also, in Figure 4(c) of the updated manuscript, how come a benign client has also backdoored data (stars) displayed?
> > My second concern is on the filtering thresholds which are critical elements to the methods but are mainly chosen heuristically. I hope there could be more systematical insights to it.   Lastly, the impact of number of surrogate dataset are not comprehensively understood. In the extreme case of 0 surrogate dataset, the problem is reduced to the original backdoor problem, therefore one would expect this number plays a positive role in affecting the defense abilities, but this trend is not seen or explained. Due to these concerns, I will remain my scores.

---

> > > ### Author Response · Authors · 2023-11-23
> > >
> > > > **Q1:** Knowing the filtering metrics, the backdoor attacker can inject this into its objectives by ensuring the model accuracy of both the surrogate dataset and backdoored task, just as a common backdoor task. Therefore it is still unclear to me how the model accuracy of surrogated dataset are still compromised by large margins in these circumstances.
> > >
> > > ***Ans for Q1):*** We agree that adaptive attacks are crucial for defense methods. Accordingly, we have conducted experiments to verify the effectiveness of our method.
> > >
> > > First, we would like to note that:
> > >
> > > - Malicious attackers who reject to follow the proposed protocol fail to mount a backdoor attack.
> > > - Adaptive attacks are considered in our experiments, demonstrating the effectiveness of our method.
> > >
> > > We appreciate your concern about adaptive attacks. Aligning with your concerns, we have conducted experiments, demonstrating the efficacy of our method. We would appreciate it if you could provide more details to implement adaptive attacks. We acknowledge time is limited, so we sincerely hope our work can inspire more adaptive attacks, aligning with your valuable comments.
> > >
> > >
> > > > **Q2:** In Figure 4(c) of the updated manuscript, how come a benign client has also backdoored data (stars) displayed?
> > >
> > > ***Ans for Q2):*** We apologize for the misunderstanding.
> > >
> > > The figure illustrates the test phase, where test samples are visualized. Namely, Figure 4 shows the feature distribution of **test data**, which includes both benign and backdoored data.
> > >
> > >
> > > > **Q3:** My second concern is on the filtering thresholds which are critical elements to the methods but are mainly chosen heuristically. I hope there could be more systematical insights to it.
> > >
> > > ***Ans for Q3):*** Thanks for your kind suggestion.
> > >
> > >
> > > > **Q4:** The impact of the number of the surrogate dataset is not comprehensively understood. In the extreme case of 0 surrogate dataset, the problem is reduced to the original backdoor problem, therefore one would expect this number plays a positive role in affecting the defense abilities, but this trend is not seen or explained.
> > >
> > > ***Ans for Q4):*** Thanks for your valuable comments.
> > >
> > > - Following your valuable comments, we will complete the results, where the performance varies with the number of surrogate data.
> > > - Our previous response mainly focuses on verifying the robustness of our method against the number of surrogate data, overlooking the mentioned extreme case.
> > > - We would like to note that the surrogate data are sampled from a pre-defined data distribution different from the natural data.

---

> ### Author Response · Authors · 2023-11-18
> **Response to Reviewer ig5G (2/3)**
>
> > **Q4:** Experimental results are limited to a small ratio of clients (12/50) and poisons (20%).
>
> ***Ans for Q4):*** We appreciate your valuable comments. Accordingly, we have conducted the following experiments and added the following discussion to our revision.
>
> - The ratio of malicious clients is set as $12/50=24\%$ larger than previous works [2,3,4]. In response to your constructive comments, we have conducted experiments using a larger ratio, $30/50=60\%$. The results are reported in Table 1-2 as below. We can see that the proposed method achieves good performance even when attackers constitute the majority.
>
> **Table 1-2**
> |Defense||Atk Num = 0|||Atk Num = 10|||Atk Num = 20|||Atk Num = 30||
> |:-:|:-:|:-:|:-:|:-:|:-:|:-:|:-:|:-:|:-:|:-:|:-:|:-:|
> ||Acc|Atk Rate|Main Acc|Acc|Atk Rate|Main Acc|Acc|Atk Rate|Main Acc|Acc|Atk Rate|Main Acc|
> |FedAvg|89.35|**0.67**|89.5|79.46|97.23|87.17|81.61|94.82|**89.3**|81.99|96.92|**89.94**|
> |RFA|87.51|1.16|87.56|80.83|81.25|87.24|80.4|86.75|87.23|79.67|91.88|86.91|
> |MultiKrum|88.33|1.22|88.4|88.32|1.35|88.47|80.35|88.31|87.3|81.03|97.01|88.88|
> |Coomed|87.91|0.96|87.91|81.41|89.1|88.57|80.94|91.51|88.27|80.64|95.46|88.3|
> |Normclip|79.71|1|79.65|73.8|83.57|79.77|73.16|87.8|79.43|73.26|89.8|79.68|
> |Nira(ours)|**89.74**|0.94|**89.78**|**88.85**|**1.24**|**88.92**|**87.67**|**2.23**|87.88|**85.13**|**3.46**|85.38|
>
> - The ratio of poisons is set as $20\%$ comparable with previous works [1,2]. In response to your constructive comments, we have conducted experiments using larger ratios, $40\%, 60\%$, and $80\%$. The results are reported in Table 1-3 as below. Our results show that our method still outperforms baseline methods and the attackers fail to enhance the attack rate by increasing the poison ratio.
>
> **Table 1-3**
> |Defense||Poison Ratio = 20%|||Poison Ratio = 40%|||Poison Ratio = 60%|||Poison Ratio = 80%||
> |:-:|:-:|:-:|:-:|:-:|:-:|:-:|:-:|:-:|:-:|:-:|:-:|:-:|
> ||Acc|Atk Rate|Main Acc|Acc|Atk Rate|Main Acc|Acc|Atk Rate|Main Acc|Acc|Atk Rate|Main Acc|
> |FedAvg|81.1|99.25|**89.16**|80.48|99.75|**88.53**|79.05|99.69|**86.95**|76.92|99.78|84.62|
> |RFA|79.44|89.51|86.71|78.12|88.55|85.07|75.95|92.21|83.03|74.95|85.54|81.54|
> |MultiKrum|80.36|99.47|88.37|79.9|99.75|87.9|77.88|99.69|85.67|74.92|99.78|82.42|
> |Coomed|80.28|98.04|88.15|79.67|99.32|87.6|78.29|99.78|86.13|75.01|99.75|82.51|
> |Normclip|71.92|92.34|78.45|69.27|93.13|75.62|66.38|93.55|72.48|61.66|93.79|67.31|
> |Nira(ours)|**84.78**|**4.34**|85.09|**84.9**|**4.58**|85.16|**85.02**|**3.92**|85.23|**84.8**|**4.4**|**85.09**|
>
> In sum, experiments with larger ratios of malicious clients and poisons further underscore the robust effectiveness of the proposed method in diverse settings. Thanks again for your constructive comments.
>
> > **Q5:** Why do the filtering thresholds vary with time in experiments? Are there any insights on how to adjust them systematically?
>
> ***Ans for Q5):*** Thanks for pointing out this potentially confusing problem.
>
> - In the federated learning scenario, the accuracy of client models on local data improves with increasing training rounds. Since the proposed method aligns the feature distributions of local data and surrogate data, the performance of client models on surrogate data also improves. Therefore, the accuracy threshold $\sigma_1$ increases and the distance threshold $\sigma_2$ decreases as the training rounds progress.
> - We introduced an efficient method in Appendix E.5 for selecting thresholds: First, train a few rounds on a small number of identified benign clients, referring to the accuracy of the model on surrogate data and the similarity of features during the training process to assist in selecting $\sigma_1$ and $\sigma_2$. Then the selected $\sigma_1$ and $\sigma_2$ are used for the training of all clients.
>
> Following your valuable comments, we have added the above discussion to our revision.
>
> > **Q6:** What does "features of the model" mean and how to determine the value of b?
>
> ***Ans for Q6):*** We apologize for the confusing description.
>
> - The "features of the model" refer to the output features of a specific layer in a neural network.
>
> Following your valuable comments, we have fixed it in our revision.
>
> - b stands for the number of noise samples used for evaluating local models. Considering that using all surrogate data will cause significant computation overhead, we set it to a fixed value of 128 for all experiments.
> - Following your valuable question, we conduct ablations on the sensitivity of b and report results in Table 1-4 as below. We can see that b has a limited impact on the performance.
>
> **Table 1-4**
> |Defense|Metric|||Noise Sample Num b|||
> |:-:|:-:|:-:|:-:|:-:|:-:|:-:|
> |||64|128|256|512|1024|
> ||Acc|87.31|87.67|**87.7**|87.54|87.24|
> |Nira|Atk Rate|1.92|2.23|2.89|**1.9**|2.69|
> ||Main Acc|87.42|**87.88**|87.83|87.69|87.49|
>
> We have added the above results and discussion to our revision.

---

> ### Author Response · Authors · 2023-11-18
> **Response to Reviewer ig5G (3/3)**
>
> > **Q7:** In Table 1, the definitions of "Acc", "Atk Rate" and "Main Acc" are not explicitly explained and therefore confusing.
>
> ***Ans for Q7):*** Thanks for pointing out this potentially confusing problem.
>
> - "Acc": the prediction accuracy computed on the clean and poisoned test samples;
> "Atk Rate": the attack success rate measuring the proportion of poisoned test samples classified as target labels by the global model;
> "Main Acc": the prediction accuracy computed on clean test samples.
> - We explained the above definitions in the first paragraph of Sec. 4 and the first paragraph in Sec. 4.2.
>
> Following your valuable comments, we have highlighted these definitions in the table's caption to avoid confusion. Thanks again for your valuable comments.
>
> > **Q8:** The experiments are conducted on relatively limited settings, i.e., only one triggered-based backdoor attack is considered, and small datasets.
>
> ***Ans for Q8):*** Thanks for your valuable comments.
> - We set the experimental settings following previous outstanding works [1,2,3].
> - We appreciate your suggestion. We would like to note that completing the experiments on the ImageNet dataset during the discussion period is challenging. Thus, we will add the mentioned large-scale datasets in our revision when we finish the experiments.
>
>
> > **Q9:** What is the ratio of the number of surrogate data and real data for each client? How will this ratio affect the performance?
>
> ***Ans for Q9):*** Thanks for pointing out the lack of information about surrogate data. We have added the following information and experiments to our revision.
>
> - The size of the surrogate dataset is $2000$, which is a small proportion of the utilized datasets, i.e., $3.33\%$ for CIFAR-10, $2.86\%$ for FMNIST, and $0.33\%$ for SVHN.
> - Following your valuable question, we conduct ablations on the sensitivity of the size of the surrogate dataset and report results in Table 1-5 as below. We can see that the size of the surrogate dataset has a limited impact on the performance. Even when the surrogate dataset size is only 500, the proposed method still demonstrates excellent performance.
>
> **Table 1-5**
> |Defense|Metric||||Surrogate Dataset Size||||
> |:-:|:-:|:-:|:-:|:-:|:-:|:-:|:-:|:-:|
> |||500|1000|2000|3000|4000|6000|8000|
> ||Acc|87.74|**87.89**|87.67|87.26|86.69|87.75|87.61|
> |Nira|Atk Rate|1.6|3.22|2.23|**1.18**|3.17|2.23|3.26|
> ||Main Acc|87.84|**88.01**|87.88|87.27|86.89|87.87|87.81|
>
> > ***Reference***
> >
> > [1] Xie, Chulin, et al. "Dba: Distributed backdoor attacks against federated learning." International conference on learning representations. 2019.
> >
> > [2] Baruch, Gilad, Moran Baruch, and Yoav Goldberg. "A little is enough: Circumventing defenses for distributed learning." Advances in Neural Information Processing Systems 32 (2019).
> >
> > [3] Zhang, Zhengming, et al. "Neurotoxin: Durable backdoors in federated learning." International Conference on Machine Learning. PMLR, 2022.
> >
> > [4] Wang, Hongyi, et al. "Attack of the tails: Yes, you really can backdoor federated learning." Advances in Neural Information Processing Systems 33 (2020): 16070-16084.

---

> ### Author Response · Authors · 2023-11-20
> **Welcome for more discussions**
>
> Dear reviewer #ig5G,
>
> Thanks for your valuable time in reviewing and constructive comments, according to which we have tried our best to answer the questions and carefully revise the paper. Here is a **summary of our response** for your convenience:
> - (1) **The proposed approach is based on strong assumptions**: We have explained that malicious attackers who reject to follow the proposed protocol fail to mount a backdoor attack.
> - (2) **Inject backdoor triggers and objectives into the surrogate dataset**: Following your constructive comments, we have conducted experiments where we inject backdoor triggers into the surrogate dataset on a malicious client. Results are shown in the given Table 1-1, which illustrates that **injecting backdoor triggers on the surrogate data will also be detected by the proposed method**.
> - (3) **The metric may be tainted**: We have explained that the proposed metric (accuracy on surrogate data) is impervious to variations in the attacker’s ratio, and the supplementary metric has the risk of being tainted. By integrating these two metrics, the proposed method achieves a more potent defense performance.
> - (4) **Small ratio of clients and poisons**: Following your constructive comments, we conducted experiments using a larger attacker ratio, $30/50=60\%$, and larger poison ratios, $40\%, 60\%$, and $80\%$. Results are shown in the given Tables 1-2 and 1-3 in responses, which further underscore the robust effectiveness of the proposed method in diverse settings.
> - (5) **Filtering thresholds issues**: We have discussed why thresholds vary with time in experiments and how to efficiently adjust them.
> - (6) **Determine the value of b**: According to your insightful question, we explained the value of $b$ used in our paper. We also conducted ablations on the sensitivity of b and report results in Table 1-4, which shows b has a limited impact on the performance.
> - (7) **The ratio of the number of surrogate data**: According to your insightful question, we explained the size of the surrogate dataset in our paper. We also conducted ablations on the sensitivity of the size of the surrogate dataset. The results shown in Table 1-5 in response illustrate that **the proposed method still demonstrates excellent performance even when the surrogate dataset size is only 500**.
>
> We humbly hope our response has addressed your concerns. If you have any additional concerns or comments that we may have missed in our responses, we would be most grateful for any further feedback from you to help us further enhance our work.
>
> Best regards
>
> Authors of #1917

---

> ### Author Response · Authors · 2023-11-21
> **Do our responses address your concerns?**
>
> Based on your constructive comments, we have tried our best to answer the questions and carefully revise the paper. We would like to verify whether the answers meet your expectations. If there is anything that hasn't been done well, we stand prepared to continue supplementing in the upcoming time. Thank you again for your valuable insights, which enhanced our work a lot. Would you mind reconsidering the rating if our responses convince you?

---

> ### Author Response · Authors · 2023-11-22
> **Look forward to your reply**
>
> Dear Reviewer ig5G,
>
> We would like to express our sincere gratitude for your insightful review. Hope our responses have successfully addressed your concerns of this paper. As the final deadline for reviewer-author discussion is approaching, we still look forward to your valuable feedback if any concerns remain. We are ready to provide further elaboration and engage in a more in-depth discussion. If you are satisfied with our replies, please don't hesitate to update your score.
>
> Best wishes,  Authors

---

### Meta-Review · Area_Chair_Wvp3 · 2023-12-11

**Metareview:**

(a) Summarize the scientific claims and findings of the paper based on your own reading and characterizations from the reviewers.

When there are backdoor attacks in federated learning, advanced backdoor defense methods involve modifying the server’s aggregation rule to filter out malicious models through some pre-defined metrics. However, calculating these metrics involves malicious models, leading to biased metrics and defense failure. A straightforward approach is to design a metric not tainted by malicious models. For instance, if the server has private data to evaluate model performance, then model performance would be an effective metric for backdoor defense. However, directly introducing data-related information may cause privacy issues. It is demonstrated that defense metrics fail when the model used is maliciously trained. A new backdoor defense is proposed, called Nira. For privacy reasons, Nira uses pure noise as data to filter out malicious clients. It is experimentally demonstrated that Nira is a good defense against backdoor attacks under federated learning scenarios.

(b) What are the strengths of the paper?

The proposed method does not need the assumption that the majority of the clients are benign. It also does not need access to an auxiliary dataset which some previous methods require. Experiments show that the proposed method performs well on three vision datasets. The paper is well-written and easy to follow in general. The proposed noise-guided defense is novel among the defense literature.

(c) What are the weaknesses of the paper? What might be missing in the submission?

It is not always true that the defense metrics fail when the model used is maliciously trained. There are defenses that work well under maliciously trained models under federated learning. The paper assumes a weak adversary who follows all the protocols but only injects some examples. Although it is true that some deviation from the protocols will reveal the identity of the adversary, characterizing and understanding such tradeoff is the key to defending against adversaries. This point needs to be properly addressed in the paper. Also, the reason for the effectiveness of noise alignment and filtering lacks clarity. It remains uncertain why attackers struggle to align their data with noise. The provided rationale for Nira's ability to defend against adaptive attacks is not clear too. The authors shows that Nira is robust to hte adaptive attack where attackers split their poisoned dataset into poisoned and benign segments, aligning only the surrogate samples with the data within the benign segments. The claimed independence to whether the adversary controls majority of clients or not is not empirically verified.

**Justification For Why Not Higher Score:**

The strong assumptions on the adversary makes it difficult to judge the robustness of the proposed defense. One of the main contribution, which is the fact that the proposed defense does not require majority to be honest, is not properly verified experimentally.  The authors claim that protocol violating adversary will be filtered out. This point is not clear. Given the vast freedom of what the adversary can do, especially in the federated scenarios, it is not clear if there exist a stronger adversary who can bypass the filter or not.

**Justification For Why Not Lower Score:**

N/A

---

### Decision · Program_Chairs · 2024-01-16

Reject